



# What makes seep carbonates ignore self-sealing and grow vertically? The role of burrowing decapod crustaceans

Jean-Philippe Blouet[1,2,3], Patrice Imbert[3,4], Sutieng Ho[3,5], Andreas Wetzel[6], and Anneleen Foubert[1]

[1]*Department of Geosciences, University of Fribourg, Chemin du Musée 6, 1700 Fribourg, Switzerland; jeanphilippe.blouet@gmail.com; anneleen.foubert@unifr.ch*

[2]*Department of Geosciences, Environment and Society, Université Libre de Bruxelles, Avenue Franklin Roosevelt 50, 1050 Bruxelles, Belgium*

[3]*Fluid Venting System Research Group, Nancy 54000, France*

[4]*Université de Pau et des Pays de l'Adour, 64000 Pau, France, patrice.imbert@univ-pau.fr*

[5]*Ocean Center, National Taiwan University, No. 1, Sec. 4, Roosevelt Road, 10671, Taipei, Taiwan, sutieng.ho@gmail.com, sutieng.ho@fluid-venting-system.org*

[6]*Department of Environmental Sciences – Geology, University of Basel, Bernoullistrasse 32, CH-4056 Basel, Switzerland, andreas.wetzel@unibas.ch*

*Correspondence to*: Jean-Philippe Blouet (*jeanphilippe.blouet@gmail.com*)

## ABSTRACT

We investigated the mechanisms that govern vertical growth of seep carbonates by studying the sedimentary architecture of a 15-m-thick, 8-m-diameter column of limestone encased in a deep-water marl succession in the Middle Callovian interval of the Terres Noires Formation in the SE France basin. The limestone body, a.k.a. "pseudobioherm" is characterized by intense bioturbation, with predominant burrows of the *Thalassinoides / Spongeliomorpha* suite, excavated by decapod crustaceans. Bioturbation is organized in three tiers. The upper tier corresponds to shallow homogenization of soupy sediment and the second one to pervasive burrowing dominated by *Thalassinoides* passively filled by pellets. Both homogenized micrite and burrow-filling pellets are depleted in $^{13}$C in the range -5 to -10 ‰. The deepest tier in contrast is filled by diagenetic cements arranged in two phases. The first cement phase makes a continuous rim coating the burrow wall, in which carbon isotope data show consistent $^{13}$C depletion near -8 ‰ to -12 ‰, indicating precipitation by anaerobic oxidation of methane in the sulfate-methane transition zone. In contrast, the second cement phase is dominated by saddle-dolomite indicating precipitation at a temperature > 80 °C, largely post-dating the burial of the pseudobioherm. The late final blocking of the burrows means that vertical fluid communication was possible over the whole thickness of the pseudobioherm up to the seabed during its active growth. Vertical growth is related to the presence of this open burrow network, providing a high density of localized bypass points across the intra-sediment calcite precipitation zone in the sulfate-methane transition zone and preventing self-sealing from blocking upward methane migration and laterally deflecting fluid flow. One key characteristic that prevented passive fill of the burrows is their geometric complexity with numerous subhorizontal segments that could trap sediment shed from shallower bioturbation tiers.

## Key Words

Seep carbonates; bioturbation; decapod crustaceans; permeability; focused fluid flow; burrow connectivity



## 1. Introduction

Seep carbonates are produced by the anaerobic oxidation of methane (AOM) or other heavier hydrocarbons coupled with seawater sulfate reduction (Boetius et al., 2000; Orcutt et al., 2010; Zwicker et al., 2018). Anaerobic hydrocarbon oxidation generates dissolved bicarbonate ions that co-precipitate with seawater calcium cations into aragonite, calcite, or dolomite. The other product of the general reaction is sulfur as $H_2S$ or HS that combine, with available iron to precipitate iron sulfides while the excess remains in solution and is potentially re-oxidized at the sea floor (Jourabchi et al., 2005; Karaca et al., 2010). In settings with active upward advection of hydrocarbons, the depth of the reaction front, the so-called sulfate-methane transition zone (SMTZ), typically lies within a few centimeters below the seafloor (Regnier et al., 2011), where a strong redox gradient favors the settlement of chemosymbiotic macrofauna (Kiel et al., 2010). Seep carbonates typically appear as concretionary bodies exhibiting a wide diversity in shape and size ranging from isolated nodules a few centimeters in diameter (e.g. Haas et al., 2010) to massive mound-shaped structures tens of meters in diameter (e.g. Kauffman, 1996). They are commonly associated with other fluid expulsion features such as pockmarks (Ho et al., 2018a, b). Oceanographic observations demonstrated that hydrocarbon seepage at the seabed is a ubiquitous phenomenon at global scale (Judd and Hovland, 2007), and seep carbonates fossilize this elusive phenomenon in the rock record. As such, seep carbonates mark the out let of underlying hydrocarbon migration pathways such as permeable layers, faults, fluid chimneys, hydraulic fractures, etc., and they have been used to reconstruct fluid-flow mechanisms in the shallow seabed sediment (Hovland, 1982; et al., 1985, 1994; Gay et al., 2003; Mazzini et al., 2003; Nyman and Nelson, 2011; Agirrezabala et al., 2013; Ho et al., 2016). Seep carbonate systems are commonly observed on seismic data as vertical or sub-vertical stacks of lenticular anomalies, which document that seep sites can be long–lived features (Kauffman et al., 1996; Hovland and Judd, 1988; Plaza-Faverola et al., 2011). In particular, Ho et al. (2012) used the morphology and vertical variation of amplitude anomalies to reconstruct qualitatively the history of fluid leakage intensity.

One mechanism that governs the growth and stacking of seep carbonate bodies is self-sealing (Hovland, 2002) whereby concretion grow that seep sites blocks vertical migration along the initial venting domain and promotes its lateral shift, leading to precipitation of laterally extensive slabs of methane-derived carbonate. Slabs of 50-150 m width identified from seafloor reflectivity in the Black Sea have been interpreted to result from self-sealing (Naudts et al., 2008). Similar patterns of lateral gas deflection have been reported in settings where gas hydrates are stable, so that their growth can block fluid migration pathways up to a scale of 5 km² (Casenave et al., 2017). At the same time, stacks of small, 20–200 m-diameter subcircular or elongate seismic amplitude patches described as "pipes" (Bünz et al., 2003; Berndt, 2005; Petersen et al., 2010; Løseth et al., 2011) or "chimneys" (Heggland, 1998; Ligtenberg, 2003; Løseth et al., 2009; Hustoft et al., 2010; Ho et al., 2016) have been described and related to hydrocarbon leakage indicators in a number of basins. Such stacks of seismic amplitude patches can exceed 1 km in thickness (Løseth et al., 2011). They highlight situations where self-sealing is restricted to patches 20 to 200 m in diameter while vertical growth predominates on a large scale (Plaza-Faverola et al., 2011). Seep carbonate bodies at all scales thus reflect a competition between lateral and vertical growth (Hovland et Judd, 1988). Self-sealing has been advocated as the dominant process leading to lateral growth, but what governs vertical growth at the scale of a seep carbonate body?

It is the purpose of the present study (1) to describe a well-exposed 15-m-thick, 8-m-diameter columnar carbonate body (initially described as a "pseudobioherm") and its sedimentary/diagenetic architecture, (2) to



interpret the cm-to-m-scale processes fostering the vertical growth/aggradation of the pseudobioherm, and (3) to replace this small-scale architecture in the frame of the permeability field in and around the pseudobioherm.

## 2. Geological Setting

The studied mid-Callovian pseudobioherm is located near the village of Aurel, about 50 km to the ESE of Valence city in SE France. During the Callovian, the area was part of the SE France basin, a ca. 150 km-diameter embayment along the northern margin of the Alpine segment of the opening Tethys Ocean (Fig.1A-B).

### 2.1. Geodynamic context

The SE France Basin resulted from the Triassic rifting of the nascent Alpine Tethys (Lemoine et al., 2000; Masini et al., 2013). Triassic transgressive deposits above the peneplaned magmatic and metamorphic Variscan basement and Permo-Carboniferous basins entrenched therein consist of shallow-water siliciclastics and evaporites. Those evaporites acted as decollement level during the structuring of the basin. Deepening of the depositional environment during the Lower Jurassic coupled with large lateral thickness variations imply

synsedimentary tilted block tectonics linked to the paroxysmal phase of rifting (Lemoine et al., 1986). More uniform deposits from the Bathonian to the Tithonian exceeding several kilometers in thickness are attributed to the onset of ocean spreading and associated thermal subsidence of the continental margin (Fig.1B-C). During Triassic and Jurassic times, the SE France basin was elongated along a NE-SW axis inherited from Variscan structures, whereas during the Cretaceous the basin narrowed and acquired an E-W elongation due to the

centripetal progradation of carbonate platforms. E-W folding of the basin, initiated during the Aptian-Albian under the N-S compressional regime induced by the Pyrenean-Provencal orogeny, led to emersion during the Santonian. From the Oligocene onward, the Alpine orogeny overprinted previous structures with an E-W compression.

### 2.2. Structural and stratigraphic setting

A 50 km long E-W oriented seismic section, crossing through the outcrop of Aurel, allowed the description of the local structural setting from the edge to the center of the basin (Fig. 1A; Roure et al., 1992; Wannesson and Bessereau, 1999). About ten kilometers east of Aurel, these authors interpreted an offset in the basement reflection as evidence for two major westward-dipping normal faults rooted in the Paleozoic

substratum. Early Jurassic tilted blocks are visible in the western part of the profile, whereas they have been reactivated as reverse structures during compressive events in the eastern part. The outcrop of Aurel is located in the middle of a large-scale domal structure (formerly known as the 'Aurel anticlinorium'; Artru, 1972)

     An exploration borehole drilled to a depth of 2800 m less than a kilometer away from the outcrop provided information on the stratigraphic column below the outcropping Callovian, down to Toarcian-Aalenian

(COPEFA, 1967) (Fig. 1C-D). The uniform Toarcian-Aalenian 'Marly Lias' has been partially penetrated for 1.5 km; the overlying 600 m thick 'Calcareous Dogger' is a limestone-marl alternation dated Bajocian to Bathonian pro parte. The Bathonian to Oxfordian Terres Noires Formation is a uniform package of shales about 1.9 km thick in the Aurel area; it grades upward into limestone-marl alternations locally known as the 'Argovian' facies (Artru, 1972).



### 2.3. Potential oil and gas sources

Potential hydrocarbon source rocks of the SE France basin have been studied in a report by Institut Français du Pétrole (IFP, 1982a, b, c) and later by Wannesson and Bessereau (1999) and Mascle and Vially (1999). In stratigraphic order, these potential source rocks are:

- Carboniferous coals in post-Hercynian basins, known from outcrops and coal mining around the SE-France Basin but speculative in the vicinity of Aurel.
- The "Liasmarneux" Formation (Marly Lias) is over 1500 m thick in the Aurel borehole and containson-average 0.6 % TOC (Wannesson and Bessereau, 1999).
- The Terres Noires Formation yielded TOC values from 0.5-1 %. Common occurrence of oil in septarian concretions indicate that some hydrocarbons were generated in the Terres Noires Formation

### 2.4. Seep carbonates in the Jurassic SE France basin

Lenticular carbonate bodies scattered in the Terres Noires Formation have been intensively studied over the past 60 years in the basin, starting with Orgeval and Zimmerman (1957). Artru and Gauthier (1966) and Artru (1972) proposed a first interpretation as sponge bioherms based on the abundance of disarticulated spicules; however, the lack of in-situ frame building organisms led Flandrin et al. (1974) to question their identification as bioherms. To account for this uncertainty, Gaillard et al. (1985) and Rolin (1987) coined the term 'pseudobioherm' (PBH), defined as "carbonate anomalies of early diagenetic origin, characterized by the predominance of endobenthic organisms". Rolin (1987), Rolin et al. (1990) and Gaillard et al. (1992) noticed similarities between the numerous chemosynthetic bivalves often associated with the PBH's (pseudobioherms) and the fauna discovered a few years earlier at cold hydrocarbon seeps (Suess et al., 1985). Since then, the hypothesis of hydrocarbon seepage has been developed at the outcrop of Beauvoisin, where 19 PBH's cluster within a circle of 800 m diameter and vertically distributed over 350 m. Gay et al. (2018) interpreted the Beauvoisin cluster as a long-lived giant pockmark and assigned the multiple generations of PBH's to self-organized lateral migration of the fluid conduit over time.

### 3. Material and Methods

The paleogeographic, structural and stratigraphic context of the Aurel PBH has been determined at regional scale based on the 1/50.000 geological map (Flandrin, 1974) and the Aurel borehole (COPEFA, 1967). At basin scale, the context has been evaluated using seismic data (Roure et al., 1992; Wannesson and Bessereau, 1999), isopach and facies maps (Debrand-Passard, 1984), and a few exploration boreholes compiled by Wannesson and Bessereau (1999).

Polished slabs of rock were studied in natural light and 365 nm ultraviolet light (UV) emitted by a Vilber VL-6 (Eberhardzell, Germany). Thirty thin sections were examined with optical microscopy using plane and cross-polarized light, cathodoluminescence (CL) and epifluorescence UV (epi-UV). The CL was generated by a CITL system (Hatfield, United Kingdom), model CCL 8200 ink4 (12kV, 450 mA). The epi-UV was



emitted by a Leica (Wetzlar, Germany) EL 6000-light mounted on a Leica DMRXP microscope (bandpass 340-380nm).

For X-ray diffraction (XRD), 27 samples were crushed manually in an agate mortar. The powders were analyzed with a Rigaku (Tokyo, Japan) Ultima IV diffractometer system equipped with a Cu X-ray tube, operated at 40 kV and 40 mA, and with a D-Tex linear detector. Scans were run from 5° to 70°2θ, with a step interval of 0.01°2θand a goniometric speed of 120 s/°2θ. The identification of all minerals was performed using the Rigaku PDXL2 software package and the ICDD Powder Diffraction File 2014 database (International Centre
for Diffraction Data).

       For stable oxygen and carbon isotope analyses, 40 samples of carbonate cements were selected. They were taken from polished blocks and on rock chips using a handheld microdrill under the alternation of natural and UV light. Due to maximum resolution of drilling bit, only thick enough cements could be sampled individually, whereas whole grainstone was taken as bulk, including cement and grains. Samples were analyzed
using a Kiel III automated carbonate preparation device coupled to a Finnigan MAT (Bremen, Germany) 252 isotope-ratio mass spectrometer. Carbonate material was reacted with 100 % phosphoric acid for 10 min at 70 °C. The produced $CO_2$was then passed through the isotope-ratio mass spectrometer for masses 44, 45, and 46 measurements alternately with the measurement of a calibrated reference $CO_2$ gas. Instrumental precision was monitored by analysis of NBS 18, NBS 19, and LSVEC reference material. Precisions are –0.05 ‰ for carbon
and –0.14 ‰ for oxygen.  Isotope results are given relative to the Peedee belemnite standard (PDB).

## 4. Results

       Four pseudobioherms have been identified within 1 km of the Aurel borehole (Artru, 1972, Fig.1D). Three of them near the Vaunière ruin are covered by vegetation and the relationships to the surrounding marls
are not visible. In contrast, the PBH at "La Touche" is well exposed in lateral continuity with its background sediments; it is the focus of this paper. The outcrop exposes an alternation of mid-Callovian marls and thin platy limestone beds dipping 15° towards ESE (log in Fig. 2). Two exposures of massive limestone are visible ca. 30 m apart along a small thalweg roughly oriented E-W. The upper one, on the right bank of the thalweg, is a 7-10 m-wide columnar body that interfingers with background marl-limestone alternations. It is exposed over a
thickness of 15 m. The top is truncated by erosion and its lower part is cut by a low-angle fault (N40, 38W) filled by a cm-thick calcite vein. The vein shows crude striae oriented N 95-N 125, without direct evidence for normal vs. reverse movement. The lower exposure is a westward-dipping subcircular section ca. 8 m in diameter on the left bank of the same thalweg. It exposes the same facies as the upper one over 6 m. The relationships between the limestone body and its host sediments are poorly exposed and its basal contact with encasing marls is not
visible. The low fault angle and its orientation, perpendicular to the trend of the exposures, suggest that these are two parts of the same columnar body offset by a late reverse fault.

### 4.1. Architecture of the pseudobioherm

       Facies mapping of the southern flank (Fig. 3A-B) and eastern flank (Fig. 3C-D) shows that the
columnar PBH is made up of stacked lenses. Individual lenses are a few tens of cm in thickness, most of them pinch out within the massive body. They are separated by cm-thick marl joints with or without nodules or merge laterally and vertically. The center of an individual lens generally consists of massive limestone, which grades



outward into nodular limestone that in turn splits laterally into several wedges that grade into nodular marl. Nodules feather out into background marl over 1-2 meters. Lenses in lateral continuity with limestone beds in the background sediments have the greatest lateral extension and are the most indurated. Overall, the PBH can be subdivided vertically into 3 units as follows, from base to top:

- Unit 1. At the base, it is partially truncated by a fault. It mostly exposes nodular limestone grading laterally into nodular marls. Locally some limestone lenses tend to be stacked vertically. The contact to Unit 2 above is defined by an abrupt apparent lateral offset by 2–3 meters westward.

- Unit 2. The lowermost bed continues laterally into a prominent limestone bed in the background deposits identified as 'marker bed A'. The middle portion of Unit 2 is made up of lenses consisting of nodular to massive limestone facies, locally organized in vertical stacks. The top of Unit 2 becomes gradually more massive. The transition from Unit 2 to Unit 3 is, once again, an apparent offset by several meters northward, similar to the transition from Unit 1 to Unit 2.

- Unit 3. The lowermost bed of Unit 3, identified as 'marker bed B', again continues laterally into a limestone bed of the host sediment and is particularly massive. It is almost entirely made up of the massive facies along the oblique section plane of the southern flank of the PBH.

### 4.2. Facies of the pseudobioherm and background sediments

### 4.2.1. Facies 1 (F1): Background, marl-limestone alternation

In the studied area, the typical middle Callovian facies consists of marls with intercalated limestone beds, a few cm to 20 cm-thick, regionally known as "Marnes à plaquettes rousses" (marl with platy, rusty weathered beds) (Artru, 1972; Flandrin, 1974) (Fig. 4A). Individual limestone beds commonly show plane bed lamination and current ripples, as well as occasional base-of-bed bioturbation (Fig. 4B). The only common macrofossils are shells of *Bositra sp.*, a bivalve typically associated to dysoxic deposits (Meesook et al., 2009). In thin sections, both the background marl and the limestone beds exhibit a dominant wackestone fabric with a similar grain content composed of microfossils and silt-grade quartz. What distinguishes them is the matrix, shaly in marls (Fig. 4C) and made of microsparite in limestone beds (probably recrystallized micrite, Fig. 4D). Laminae in limestone beds appear as mm-scale alternations of wackestone and packstone (Fig. 4E). The most abundant microfossils are spheres with a diameter of ca. 100 μm, likely calcispheres and/or recrystallized radiolarians, and sponge spicules (Fig. 4F). Echinoderm ossicles are rare. Most microfossils have been recrystallized into sparite. Sinuous filaments, up to several hundreds of μm in length for a width of less than 10 μm correspond to the structures identified by Negra et al. (2011) as fragments of exfoliated shells of *Bositra sp*. (Fig. 4C).

### 4.2.2. Facies 2 (F2): Nodular marl

This facies comprises the transition between background sediments and the PBH limestone body (Fig. 5A). It consists of limestone micrite nodules embedded in a marly matrix exhibiting a matrix (marl)-supported arrangement. The most common individual nodules are isolated prolate ellipsoids with the long axis parallel to the stratification. The nodules are 1–2 cm in diameter and up to 5 cm in length. Some show rod-like, straight, or curved protuberances (Fig. 5B). Nodule concentration typically increases towards the limestone body. In areas of higher concentration, adjacent nodules commonly aggregate into peanut-shaped clusters of two or three. In thin



sections, nodules show a wackestone microstructure with the same type of grains as in the previously described encasing marls and limestone beds, i.e. microfossils and detrital silt. Peloids being a key component of facies F4, as described in section 4.2.4 hereafter, we paid particular attention to the possible presence of peloids or altered

peloids in the microsparitic matrix. Neither direct observation, nor fluorescence or cathodoluminescence provided indications of the presence or former presence of peloids in F2. Nodules are weakly fluorescent under UV light (Figs. 5C-D).

### 4.2.3. Facies 3 (F3): Nodular limestone

The concentration of nodules increases towards the axis of the PBH and individual nodules may

amalgamate into nodular limestone. In that case, the matrix between the nodules is strongly indurated and shows pervasive rusty speckles (Fig. 6A).As in F2, nodules are weakly fluorescent in a non-fluorescent matrix (Fig. 6B). In contrast to nodular marl, the nodular limestone appears nodule-supported. A threshold of ~50 % nodules marks the limit between F2 that is sensitive to weathering as being a nodular marl and F3 that remains rather massive and less prone to weathering. The different weatherability of the two facies was used for PBH

facies mapping of hardly accessible zones like overhangs (Fig. 3). Thin sections show that the rusty speckles observed macroscopically consist of euhedral crystals of saddle dolomite about 100 µm in size, locally replacing and overgrowing carbonate bioclasts (Fig. 6C). The rusty color of the crystals is due to oxidation of iron along cleavage planes and gives the matrix its speckled macroscopic appearance.

### 4.2.4. Facies 4 (F4): Massive limestone


#### 4.2.4.1. Macroscopic observations and macrofauna

Facies F4 consists of grey to brownish limestone that typically interfingers with the nodular limestone. In contrast to the background sediments, no primary sedimentary structures or laminations have been observed in F4, although laminated limestone beds of the surrounding background sediment are in physical continuity with

prominent lenses of the PBH. Weathering hinders observation of facies details in the field, except on a few isolated surfaces. One of these, ca. 30 cm x 70 cm, exposes the feathering out ofF4 into marker bed A that extends farthest away from the axis of the PBH (Fig. 7A). The surface shows three articulated bivalves (Fig. 7B–D). In addition, a single, 6 cm-long articulated specimen found on the margin of the PBH was identified as a lucinid similar to 'Sp.1' described in the Beauvoisin outcrop by Rolin et al. (1990; Fig7E-F). A few small

gastropods were also observed in the outcrop and in cut samples (Fig 7G), indicating that benthic mollusks were relatively common and diversified in F4.

#### 4.2.4.2. Fabric and bioturbation

The fabricis best characterized by using combined plane light and UV light on cut sections (Fig. 8A-C). Macroscopic observation evidences a complex mosaic of patches, which observation under UV light shows

consist of two main textures: low-fluorescence micrite and granular material with various levels of fluorescence. Hand lens examination indicates that the granular limestone consists of peloid grainstone, fluorescence coming from the cement. The respective proportions of micrite and granular limestone are variable, with the former in the range 20-50 % and the latter representing 80-50 % of the total. Near the transition to facies F3, samples richer in micrite (ca. 50 %) show it as the continuous phase, with granular material defining a network of 2-3 cm

diameter patches surrounded by a darker, commonly fluorescent lining. A similar dark lining encloses 0.5-1.5-



cm diameter circular or elliptical patches variably filled by concentric cements or the association of peloid grainstone in the lower part and cements above. This texture evidences pervasive bioturbation, with two populations of burrows. Two-to-three cm diameter granular patches will hereafter be referred to as "large burrows" and 0.5-1.5-cm patches as "medium burrows". Examination under UV light (Fig. 8C) indicates that
medium burrows dominantly cut through large burrows. In addition, numerous mm-diameter burrows ("small burrows") filled by fluorescent cements are scattered over the section without clear relation to the previous two sets.

### a) Large burrows

The margins of large burrows, as defined by their dark lining, are rather irregular. This geometry suggests that large burrows make a complex branched network. They are filled with sediment, either peloid grainstone identical to that of the background indicating passive fill or finer-grained material exhibiting concentric bioclast orientation and indicating active fill, e.g. in the darker halo around T3, which shows concentric bioclast orientation.


### b) Medium burrows

The morphology of medium burrows is best characterized by comparing 3-mm-spaced parallel sections provided by the two sides of the same sawcut, thereby giving an insight into some of their 3D characteristics. In the representative sample shown in Fig. 8A and 8B, six individual medium burrow shave been followed across
the sawcut, numbered B1 to B6; B1 and B3 show an abrupt change in direction, from sawcut-perpendicular (circular section in Fig. 8A) to sawcut-parallel on the other side. BurrowB2 shows a diameter change from 0.5 to 1.5 cm over the same distance. The sketch in Figure 8D combines the observations made on the two parallel sections. Close examination of burrow B6 (Fig. 8E) shows in the upper part of the section a dark lining around the cement fill (larger white arrows). This lining can be followed into the sediment, defining a subcircular shape
whose lower part is sediment-filled and the upper part cement-filled. Burrows B4, B5 and to a lesser extent B3 also have a two-phase infill, with grainy sediment in the bottom part and cements above. The contact between intra-burrow sediment and overlying cement is generally planar, with an apparent dip in the 0°-30° range (see B5, B6 in Fig. 8B), indicating partial passive fill. Some burrows, like B6, show additional linings within the filling sediment, indicating multiphase passive infill. Most medium burrows macroscopically show a concentric
two-part cement infill of the lumen, with a ca. 1 mm-thick tan, translucent outer rim that is continuous around the lumen and a white to yellow/gold final axial fill (Fig. 8A-B). The outer rim is brightly fluorescent, in contrast with the low-fluorescence final fill (Fig. 8C). Burrow margins are commonly irregular, without it being clear whether the irregularities are biological (scratches related to burrowing in stiff sediment), chemical (related to corrosion by fluids circulating in the burrows and changing redox conditions) or possibly mechanical (local
collapse of top of burrow sediments). Overall, based on the observation of 10sections comparable in size to that shown in Fig. 8, the average distance between adjacent medium burrows ranges from 2-10 cm with an estimated average at 5 cm. Burrows show subvertical, oblique and subhorizontal segments, without preferential orientation. Combined together, the abrupt changes of orientation and diameter over short distances and the coexistence of vertical, oblique, and horizontal segments are diagnostic of burrows of the *Thalassinoides – Spongeliomorpha*
suite. These are produced by decapod crustaceans that may penetrate as deep as two meters or more below



seafloor (e.g., Sarnthein, 1972; Pemberton and Buckley, 1976). *Thalassinoides* corresponds to smooth-walled burrows excavated in sediment stiff enough to remain open and *Spongeliomorpha* to burrows scratched by body appendages of the burrower, indicating firm sediment (Wetzel and Uchman, 1998).

**c)  Small burrows**

Cement-filled small burrows (circled in red in Fig. 8A-C) are better visible under UV light. Over the 10 sections observed, a minority of these small burrows appear connected to medium burrows (e.g. t1 in Fig. 8A-C) while most appear isolated, like t2 and t3 in Fig. 8A-C, although out-of-plane contact with medium burrows cannot be ruled out. Their generally smooth wall character likely indicates *Trypanites*, i.e. borings in hard
sediment.

### 4.3. Relative timing and tiering

Based on cross-cutting relationships, it is possible to distinguish two successive phases of burrowing: a shallower tier of abundant bioturbation dominated by large *Thalassinoides / Spongeliomorpha* and a deeper tier
with scarcer burrowing limited to smaller *Thalassinoides / Spongeliomorpha* with occasional *Trypanites* boring from their cemented margin. In addition, the non-fluorescent wackestone of the intraclasts has an overall similar grain content to background sediment of facies 1 or to the nodules of facies F2-3. In contrast with background sediment, it never shows any lamination, nor layering on the scale of limestone / marl alternations in the background, while the sedimentary processes that produced the physical layering and laminations must have
been active in the PBH like in the surroundings. The homogeneous texture of the wackestone consequently reflects mixing in the uppermost soft sediment by near-surface burrowing organisms producing so-called biodeformational structures (e.g., Schäfer, 1956; Wetzel, 1991). It is thus possible to distinguish three phases of biological activity, reflecting biodeformation / bioturbation in three main tiers: homogenization of soft sediment into wackestone in phase 1/tier 1, burrowing of firm sediment by dominant large decapod crustaceans and
burrow filling by peloids in phase 2/tier 2, and burrowing of stiff sediment by smaller decapod crustaceans with occasional subsequent *Trypanites* borings in phase 3/tier 3.

### 4.4. Texture of grainstone-rich zones

Compared with previously described micrite-rich samples, zones richer in granular material (in the 70-
80 % range of peloid grainstone) show a more complex arrangement of micrite and granular patches (Fig. 9A-B). The latter group shows highly variable levels of fluorescence, with a characteristic diameter of 0.2-2 cm for homogeneous patches. Generally speaking, lower fluorescence material appears as forming intraclasts, rounded to angular, in a more fluorescent background. In the light of observations made on micrite-rich zones, the fabric appears to reflect a higher degree of bioturbation by a variety of organisms, micrite intraclasts representing in-
situ or collapsed remnants of the initial phase of homogenization (tier 1) enclosed in successive generations of burrow-filling peloid grainstone (result of tier 2 activity). In such grainstone-rich zones, the effects of phase-1 homogenization are only visible in intraclasts.

### 4.5. Microfacies and diagenesis



*Tier 1: micrite intraclasts*

Micrite homogenized by biodeformation is only found as isolated remnants between burrows of later bioturbation phases. The corresponding microfacies is thus described from intraclasts floating in the dominant peloid grainstone (Fig. 10A-B). In thin section view, these intraclasts consist of shaly-silty-bioclastic wackestone, with the same grain types as observed in background sediment (Fig. 4C), i.e. calcispheres, sponge spicules and bivalve fragments (arrows in Fig. 10C) for the bioclastic fraction.

*Tier 2: large burrows and their infill*

Burrows of tier 2 / phase 2 are filled with peloid grainstone. Peloids have a narrow size distribution in the fine-to-medium sand range, typically 200-300 μm (Fig. 10D). Their morphology is ovoid, and their internal fabric is identical to that of intraclasts of F4 and nodules of F2-3, consisting of silt / microfossil wackestone. Epifluorescence microscopy confirms previous macroscopic observations that fluorescence mainly comes from microsparite cementing the grainstone (Fig. 10E)

*Tier 3: medium burrows and their infill*

In this section and other sections dealing with the details of the concentric fill of burrows by cements, we take as a reference the center of the burrow, so that "above", "below" and similar terms mean "closer to the burrow axis" or "farther away from the burrow axis", respectively. One prominent feature of medium burrows is the presence of a ~100 μm-thick fluorescent rim around the final cement fill. Comparison of UV and natural light close-up views show that the highest levels of fluorescence come from the milky white cement highlighted with yellow arrows in Fig. 10A-C and the adjacent clear sparite. This section examination shows that this white cement consists of a high-relief light brown mineral (Fig. 10C; Fig. 11A), which in cross-polarized light shows the typical texture of flamboyant chalcedony (Sander and Black, 1988; Fig. 11B). This mineral, noted Chal-2 generally occurs in association with saddle dolomite (Dol-1). Flamboyant chalcedony commonly cuts across single sparite crystals (Fig. 11C, Chal-2 cutting in two a single crystal of Spar-1). It locally replaces older, rare occurrences of cryptocrystalline botryoidal chalcedony (Chal-1, Fig. 11 E-F). Below the fluorescent rim, discontinuous Chal-2/Dol-1 patches commonly replace the upper part of sediment/cement sequences thereby defining a complex, digitated surface of replacement. Silica replacement never occurs above a surface characterized by the occurrence of euhedral quartz crystals but can locally affect any of the underlying cements, with the exception of those cementing grainstone porosity. We refer to this surface as the "main silicification surface" (MSS), which can be traced as the envelope of euhedral quartz crystals (blue zigzag lines in Figs. 11 D-F).

Cements below the MSS will now be described inward from the dark lining that defines burrow boundaries (bold arrows in Fig. 8 E). Cements composing the dark lining are commonly interlayered with sediment, i.e. are coeval with passive burrow infill. Thin section examination indicates that dark linings show the same succession of cements irrespective of their position, intra-sediment or at sediment/cement boundary (Fig. 10A-C). It starts with a sediment-coating,0.1-0.3 mm-thick layer of brown microsparite (BMSpar, orange arrows in Figs. 10A-C) commonly covered by 0.2-0.5 mm-thick clear microsparite (CMSpar, Fig. 12A). CMSpar in turn is locally covered by sets of radiaxial grey carbonate crystals (RAx in Fig. 11D). Intra-burrow sediment can be



deposited in geopetal position above this succession, in some cases covered again by a similar BMSpar- CMSpar doublet. The sediment-BMSpar-CMSpar succession has been observed up to 3 times (Fig. 12A). Individual microspar crystals are continuous across the brown / clear boundary (Fig. 12B-C). High magnification views
indicate that the brown color of BMSpar is due to bush-forming sets of brown filaments (Fig. 12D) whose diameter is close to the limit of optical resolution (ca. 1 μm). Individual "bushes" are about 100 μm in diameter and have a smooth, convex margin. Synsedimentary cements are thus arranged into repetitive sequences made of sediment/BMSpar/CMSpar, occasionally capped by radiaxial sets of calcite crystals; these will be referred to as "sediment-cement sequences". Above the last sediment-cement sequence, the dominant cement is sparry calcite
(Spar-1), with the occasional presence of botryoidal chalcedony (Figs. 11E-F), both being locally affected by silica / saddle dolomite replacement. Cements above the MSS are dominantly sparry calcite (Spar-2) and/or saddle dolomite (Dol-2), in variable proportions Subordinate minerals include fracture-filling microcrystalline chalcedony and occasional sulfates (barite-celestite mixtures). Comparisons between plane polarized light and UV-epifluorescence images show that neither the abundant inclusions that characterize BMSpar, nor the calcite
cementing the filaments are fluorescent, in sharp contrast with CMSpar that shows both general mineral fluorescence and brightly fluorescent fluid inclusions (Fig. 12E-H). Fluorescent inclusions appear most abundant along the contact surface between BMSpar and CMSpar (Fig. 12F).

### 4.6. Stable isotopes

Delta $^{13}$C and $\delta^{18}$O measurement are tabulated in Appendix 1 and plotted in Figure 13A. The stable carbon and oxygen isotope data of the studied carbonates exhibit a trend between the marl sample with a $\delta^{13}$C of 1 ‰ and a $\delta^{18}$O of -2.2 ‰ and a pole with $\delta^{13}$C = -10 ‰ PDB and $\delta^{18}$O = -1.5 ‰ PDB. Samples of this $^{13}$C-depleted pole comprise the microsparite cements of the sediment-cement sequences (BMSpar and CMSpar) and Spar-1, along with some peloids and some nodules. Figure 13B and its close-up in Fig. 13C, based on
burrows like that shown in Fig. 10A, put these measurements in the frame of the facies / cement relationships. The typical microfacies / cement arrangement of facies 4 records the tiering of bioturbation, with cement-filled tier 3 burrows cutting across a mixed background with both tier 2 burrows and intraclasts homogenized by tier 1 biodeformation (Fig. 13B). While tier 2 burrows are uniformly filled by peloid grainstone, tier 3 burrows show a combination of passive sediment fill in the lower part and diagenetic cements filling the residual void. Figure
13C summarizes the details of this architecture and illustrates with black boxes the typical composition of samples selected for isotope analysis, in relation with the size of the sampling tool. It appears that all cements that post-date the MSS show limited $^{13}$C depletion, whereas those pre-dating it follow a mixing trend between the reference marine sediment (F1 marl) and those of pre-MSS cements that could be sampled in isolation. The values of $\delta^{18}$O range between -1 ‰ and -3 ‰ for all samples but one, a sample of Dol 1 with $\delta^{18}$O = -7.5 ‰.


### 5. Discussion

The main lines of evidence to identify seep carbonates in the fossil record are the presence of specific methanotrophic fauna (e.g. Kiel, 2010) and/or the carbon isotope signature of the carbonate minerals, generally considered diagnostic when $^{13}$C depletion exceeds -25 ‰. In Aurel, no sample meets the latter criterion.
Similarly, the scarcity of CH$_4$-related macrofauna with one single specimen of lucinid does not provide a



compelling proof for the nature of the whole carbonate body. The most convincing evidence comes from combined analysis of bioturbation, early diagenesis in and around burrows and isotope geochemistry.

### 5.1. Nature of the pseudobioherm

The co-occurrence of peloids and pervasive bioturbation strongly suggests a genetic link. In addition, the uniform size and regular ovoid morphology of the peloids, their internal grain content identical to that of intraclasts and nodules around match all the criteria that would be expected from *in-situ* recycling of local sediment by deposit feeders. We thus interpret the peloids as fecal pellets, produced either by the organisms that dug Phase-2 burrows, or close to/at the seabed and passively shed down the burrows by bottom currents, e.g.

currents that made ripples in background sediment.

    The PBH, hence, appears as an oasis of thriving benthic life, essentially endobenthic, within a domain marked by much lower abundance of benthos and scarce bioturbation. One key factor to maintain benthos development is the availability of food / energy, the photosynthetic food chain being dominant in the photic zone. Below the photic zone, benthos development depends on the availability of sedimentary organic matter

derived from the photic zone and/or continents and brought in by currents, or in a totally separate way by chemosynthesis. Very localized spots of sustained benthic life like the Aurel PBH (< 10 m in diameter) are found in particular at seep sites (e.g., Jensen et al., 1992). The Aurel PBH contains neither frame-building nor frame-binding organisms. The thickness of the PBH of at least 15 m largely exceeds the maximum penetration depth of burrowing organisms. In F4, the final fill of burrows generally consists of saddle dolomite, which is

known to precipitate from saline fluids at temperatures in the range 80-160 °C (Spötl and Pitman, 1998). The network of Phase 3burrows, thus, remained open until this temperature value was reached supposedly at a few kilometers' burial depth. In early stages of burial, i.e. during growth of the PBH, the fine-grained background sediments had a strongly anisotropic permeability, as follows: the overall vertical permeability was low due to the dominance of marls whereas lateral permeability was high due to the presence of the interbedded limestone

beds providing pathways for lateral fluid migration. In contrast, the absence of laminated limestone beds in the PBH and its pervasive homogenization during the very first stages of burial mean that the PBH sediment had an isotropic low permeability during its growth. In contrast, the open burrow network of F4 provided localized natural conduits for fluid flow during/due to compaction, making the whole PBH an effective vertical drain at large scale. We therefore interpret the vertical development of the Aurel PBH to have resulted from dominantly

vertical fluid seepage, making it a fluid seep carbonate body.

### 5.2. Nature of the fluid

    Stable carbon isotope analysis is the most commonly used method to elucidate the origin of fluids from which seep carbonates are derived (Campbell, 2006). Campbell (2006, her Fig. 9A) considers as a "classical seep

carbonate" signature a combined range of -60 to -30 ‰ for $\delta^{13}$C and -2 to +8 ‰ for $\delta^{18}$O.There appears to be a consensus that any authigenic carbonate exhibiting $\delta^{13}$C values lower than -30 ‰ can be considered diagnostic of methane oxidation, where as less depleted values may result from mixing of bicarbonate derived from other sources, in particular seawater (Campbell, 2006). In the Aurel PBH, the most $^{13}$C-depleted samples having about -10 ‰ PDB are those predating the emplacement of the main silicification surface (MSS) and occurring in

patches large enough to be sampled separately. Samples comprising both sedimentary carbonate grains and





diagenetic cement, such as micrite from intraclasts or nodules and pellet grainstone, have $\delta^{13}C$ values along the trend between normal marine sediments and the most depleted cements (Fig. 13 A). This overall pattern is consistent with a mixing trend where the signature of sedimentary marine carbonate like micrite and microfossils is partly overprinted by $^{13}C$-depleted early diagenetic carbonate cements.

The most $^{13}C$-depleted cements in this study remain significantly less depleted than the commonly accepted diagnostic -30 ‰ threshold, however. That threshold results from the successive changes in $^{13}C$ depletion undergone during the maturation of organic matter into hydrocarbons and eventual precipitation of carbonate when anaerobic oxidation of these hydrocarbons took place. The $\delta^{13}C$ of sedimentary organic matter ranges from -10 to -35 ‰ PDB, with an average of -25 ‰ PDB in marine environment. During burial, microbial

methanogenesis increases the depletion. Microbial $CH_4$ generation leads to methane with $^{13}C$ depletion from -120 to -50 ‰ PDB (Botz et al. 1996, Whiticar, 1999), while thermogenic maturation yields methane with $\delta^{13}C$ values from -50 to -35 ‰ PDB (Fuex, 1977). A comparison between carbon isotopy of recent seep carbonates with that of the gas bubbling from the seep sites evidenced a further reduction of $^{13}C$ depletion by an average ca. 20 ‰ upon carbonate precipitation (Peckmann and Thiel, 2004), which the authors interpreted to result from

mixing with dissolved inorganic carbon. Combining the successive phases of depletion / enrichment, seep carbonates should thus have typical depletion values of -100 to -35 ‰ when derived from microbial methane and -35 to -15 ‰ when sourced from thermogenic gas. A recent review of methane carbon isotopy based on more than 20,000 samples (Milkov and Etiope, 2018) indicates that the $\delta^{13}C$ of thermogenic methane ranges from -65 ‰ to -25 ‰ and very much depends on the maturity of $CH_4$, as follows: oil-associated methane overall

matches the values proposed by Fuex (1977), while early mature values are in the -65‰ to -50 ‰ range and late mature samples cluster around -30 ‰. Assuming that the average mixing effect with seawater is in the range of 20 ‰ as observed by Peckmann and Thiel (2004), we interpret the $\delta^{13}C$ of -10 ‰ observed in early cements of the Aurel PBH to indicate that the PBH seep carbonates result from anaerobic oxidation of late thermogenic methane.

Oxygen isotopes are much less consistent, with a very narrow variation range (-1.8 ‰ to -2.2 ‰) between early cements precipitated at seabed temperature and most samples of saddle dolomite precipitated at 90 °C at least (Spötl and Pitman, 1998). Moreover, one sample of saddle dolomite strongly departs from the other five with a value of -7.5 ‰, without any discernible logic. There are two possible explanations here: $\delta^{18}O$ could have been affected by exchanges with pore water over its burial history, or by meteoric water during the recent

exhumation of the PBH although we avoided sampling weathered parts of the samples. In any case, we consider the $\delta^{18}O$ signature in the PBH to be too severely affected by overprinting and did not use it any further. We therefore interpret the Aurel PBH as a column of seep carbonate precipitated at the top of a stationary 5-10 m-diameter fluid chimney supplying late mature thermogenic methane.

**5.3. Facies interpretation**

**5.3.1. Facies 1-3**

Background sediment records hemipelagic settling of fine-grained siliciclastic and carbonate particles forming marl and episodic depositional events that transported material reworked higher up the slope or on the shelf downward by density currents, e.g. dilute turbidity currents. Hydrocarbon seeping up through the limestone

column does not appear to have affected sedimentation or early diagenesis more than a few m away from the





venting site. Diagenesis of F1 is restricted to the normal basin-wide effects of burial compaction. For F2 and F3, the nodules showing an AOM-influenced isotope signature reflect an increase of AOM-generated bicarbonate to levels above the precipitation threshold of calcite, the more as the nodules become more abundant and coalesce closer to the axis of the PBH (Fig. 14). Calcite precipitation occurred in the SMTZ that has been shallower above

the PBH than in the surrounding 'background' area due to the increased methane supply (Fig. 14, cf. Paull and Ussler, 2008). In F3, the saddle dolomite crystals in the matrix between coalescing nodules indicate percolation of late diagenetic fluids from the limestone column, still acting as a local fluid conduit even if burial depth was sufficient to reach a temperature of 80°C at least.

**5.3.2. Facies 4**

Carbon isotope analysis of the infill of phase-3 burrows indicates that the main silicification surface separates an AOM-influenced early diagenesis form a late diagenesis in which saddle dolomite precipitation has a prominent role. Anaerobic oxidation of methane combined with sulfate reduction can only occur where sulfate is available, i.e. in the first few meters below seabed at the deepest. In contrast, the high temperatures required

for saddle dolomite precipitation means that late diagenesis took place at depths largely exceeding 1 km, more likely 2km assuming realistic geothermal gradients.

Most of the fabric of F4 was acquired during the early, AOM-dominated history with the late phase only affecting phase-3 burrows. The fabric actually records interplay between sedimentation, bioturbation and early diagenesis. Sedimentation at the seep must have been identical to that recorded by background facies F1 in the

first place, since hemipelagic settling and bottom current deposition act on a larger scale than the 10-m diameter of the PBH. Homogenization by biodeformation was then the first process that modified this initial soupy deposit, above the earliest cementation that started stiffening the sediment. Bioturbation phases 2 and 3 then occurred after enough MDAC had been precipitated to maintain burrows open and preservable (Wetzel and Uchman, 1998), and the presence of open burrows in turn guided upward methane migration and promoted

AOM and gradual cementation around the burrows (Wetzel, 2013). Leaving aside the effects of late silicification, the cement infill of Phase 3 burrows and their host sediment is then quite simple: what is recorded is AOM-related calcite precipitation in the SMTZ, expressed by micrite-scale crystals in the porous network of homogenized micrite, microspar-size crystals in the porous network of pellet grainstone and in the first 0.1-0.5 mm above burrow walls (BMSpar-CMSpar) and spar-size crystals (Spar-1) in the remaining free space of the

lumen. The radiaxial sets of grey carbonate crystals that occasionally terminate the sequence of cements (RAx) could be recrystallized equivalents to the aragonite fans described by Blouet et al. (2017) in the Panoche Hills (CA, USA) or by Peckmann et al. (1999) in Beauvoisin. Calcite crystal continuity across the BMSpar-CMSpar contacts indicates that the brown color simply reflects the inclusion of the filamental bushes into the growing microspar. These bushes are reminiscent of the dumbbell-shaped crystal aggregates reported by Peckmann et al.

(1999) in the Miocene seep carbonates of Marmorito (Italy), and are most likely filamentous, bush-shaped microbial colonies having grown over free surfaces of micrite intraclasts or pellets on burrow walls. The presence or absence of intra-burrow geopetal sediment would simply reflect the capacity of a burrow segment to trap pellets avalanching from above. Pellets could fall down due to mechanical burrow wall collapse, biological activity above or seabed currents inducing intra-burrow circulation. The variability of $^{13}$C depletion in micrite

(nodules and intraclasts) reflects variations in the amount of marine carbonate mixed with the AOM-derived


cement. In multi-phase samples, e.g. pellet grainstone mixing pellets with their AOM-derived cement, the variability can be related to variations between pellets (themselves derived from micrite with a variable signature) and variations in the proportion of cement in the sample (see local variability of grain / matrix proportions in patches labeled "PG" in. Fig. 10 C).

The cm-scale geometry of the SMTZ was most likely complex, with pervasive downward digitations following at any time the open burrow network that provided connection to the sulfate-bearing open water. In addition to this complex static geometry, tube ventilation by seawater is known to be induced by bottom currents ("passive ventilation" e.g. by tidal currents: Wetzel et al., 2014; Gingras and Zonneveld, 2015; Rodríguez-Tovar et al., 2019) and by the burrowing organism itself ("active ventilation"). Active ventilation has been shown to be

used by decapod crustaceans to promote microbial growth on burrow walls (Savrda, 2007). The microbial colonies of BMSpar may result one or the other type of ventilation.

     The key information provided by late diagenesis about the vertical growth of the PBH, is that the porous network made by burrow lumina remained open long after the PBH was buried, a fortiori as long as the PBH was growing. One issue of interest is silica diagenesis, which occurred both in the early, AOM-related phase of

diagenesis with precipitation of botryoidal chalcedony and more pervasively during late diagenesis. Silica is ubiquitous in the host formation as sponge spicules, so that it was readily available within and all around the PBH at all times. The early chalcedony botryoids could be due to AOM-related carbonate precipitation leading to relative enrichment of $Mg^{2+}$ ions in the open burrows, itself fostering silica precipitation (Williams et al., 1985a-b). We hypothesize that the main silicification phase recorded by the MSS may correspond to a specific

step of silica diagenesis affecting the sponge-rich interval of the PBH, perhaps upon entering the Opal C-T-Quartz transformation window around 55-85 °C (Keller and Isaacs, 1985), which overlaps with the lower range of precipitation temperature for the associated saddle dolomite.

### 5.3.3. Significance of the repeated microfacies sequence

We have interpreted sediment-cement alternations as the result of biological activity, in particular that of decapod crustaceans. The repeated succession of early AOM-related cements in Aurel is in many respects comparable to that of seep carbonates of the Moreno Formation (Paleocene, Great Valley Basin, California) that records pulsed methane supply from a deep reservoir (Blouet et al., 2017). In fact, the succession of cements in the Aurel PBH shows, like in the Moreno Formation, an evolution from smaller (microsparite) to larger (sparite)

crystals and presence of radiaxial sets of crystals possibly corresponding to the fans of aragonite needles of the Moreno Formation (Blouet et al., 2017), and from the Terres Noires Formation of Beauvoisin (Peckmann et al.,1999). The interpretation of the cement succession in the Moreno Formation was also based on the presence of corrosion surfaces that indicate episodic changes in fluid chemistry or temperature, which was not observed in Aurel. Although methane expulsion is most likely episodic on several time scales in Aurel, as it is in most

hydrocarbon seepage types (Roberts and Carney, 1997; Deville and Guerlais, 2009; Kopf, 2002; Mazzini et al., 2009; Ho et al., 2012; Jatiault et al., 2019a, b; Plaza-Faverola et al., 2011, 2015), there is little evidence in Aurel for such episodicity in sediment-cement sequences of the Aurel PBH, and the repetitive character of the sediment-cement sequence is more likely to reflect random avalanching of sediment punctuating continuous growth of microsparite cement.






### 5.4. Parameters controlling the stacking pattern

The Aurel seep carbonate body comprises vertically stacked carbonate lenses in marly intervals and shows some lateral shift across the thickest limestone event beds (marker beds A and B). Lateral deflection of upward-seeping hydrocarbons by the seep carbonates results in "self-sealing" as conceptualized by Hovland (2002). Self-sealing has been interpreted from oceanographic data in the Black Sea (Naudts et al., 2008) and in outcrop by Agirrezabala et al. (2013). The small-scale process of vertical gas migration is blocked by precipitation of seep carbonates, leading to a downward growth of a concretionary crust as evidenced by Bayon et al. (2009) using U/Th age dating. The offset of the seep carbonate column upon crossing marker beds A and B is interpreted to result from seep carbonate cementation in these beds starting from the top shallow within the seafloor. No lateral shift of the limestone body, however, has been observed when crossing thinner limestone beds. Based on the observation that massive limestone in the PBH underwent homogenization by bioturbation in the very shallow, 0–10 cm subsurface (Wetzel, 1981), we suspect that bioturbational mixing destroyed the continuity of thin limestone beds by displacing grains into the surrounding marly mud, whereas thick enough beds would just show blurred contact at the top limestone / marl contact but preserve a permeable lower part of the bed, along which incoming fluids may have migrated upslope. Upslope migration of hydrocarbon-charged fluids, in turn, shifts the zone of maximum hydrocarbon concentration laterally upslope and also the habitat of burrowers feeding on chemosymbiotic microbial communities along their burrow walls and on the seafloor.

What are now the parameters that control vertical growth of the PBH in sections where background marls only have <5 cm interbedded limestone beds? The role played by *Thalassinoides* burrows in focusing methane migration has been identified by Wiese et al. (2015) and Blouet et al. (2017). This study helps understand the specific factors that make it possible for this type of burrows to remain open in spite of likely overprinting by shallower burrow tiers. Individual phase 3 burrows that remained open until burial reached at the very least 1 km provide obvious fluid migration pathways as deep as the reach of burrowing organisms, likely in the range 1-3 m from studies of present-day decapod crustacean burrows. However, the shallow part of the burrows, within the range of action of shallow sediment homogenizers of tier 1 (ca. 10 cm below seabed?) must have been obliterated after the burrow was abandoned, cutting connection to the surface. Burrowers of tier-2 in the same way would have intersected abandoned tier-3 burrows and shed sediment and pellets down the intersected shafts. The estimated average spacing of ~ 5 cm observed between tier-3 burrows implies a burrow density of about 400 per m², in the mid-to-high density range defined by D'Andrea and DeWitt (2009), while the burrows have a complex branching geometry made of segments with highly variable azimuth and inclination. As a result, there is room for many intersections between active and abandoned phase 3 burrows. *Thalassinoides*/*Spongeliomorpha* type burrows generally show variable orientation on a cm-scale (e.g., Griffis and Suchanek, 1991; Dworschak and de Rodrigues, 1997; Stamhuis et al., 1997), as observed in the Aurel samples (Fig. 15A, adapted from Ziebis et al., 1996). The presence of low-inclination segments thus provides a dead end for sediment avalanching down from upper tiers: as soon as sediment shed from above reaches a segment whose dip is less than the repose angle of granular material, passive fill is blocked and the deeper open network is preserved for good (Fig. 15B). The geometric specificity of tier-3 burrows with subhorizontal segments thus appears as a key factor ensuring that these burrows provide an open tube network over the whole thickness of the PBH as long as there were active organisms providing the final opening to the seabed.



Vertical growth is a direct consequence of the existence of this fluid circulation highway: compaction fluids and buoyant hydrocarbons, once they reached the base of the PBH, were funneled along the burrow network up to the seabed where MDAC precipitation occurred. Once initiated, vertical growth became a self-sustained process until thick limestone beds temporarily provided an effective lateral leakage pathway, causing the episodic lateral shift observed upon crossing the marker beds.


At any time during active seep carbonate growth, the presence of occupied tier-3 burrows provided communication with open sea water on the one hand and brought up methane on the other, ensuring both MDAC precipitation in the shallow subseabed and multiple bypass pathways preventing self-sealing. Furthermore, sediment stiffness  prevented burrow collapse. The specific behavior of decapod crustaceans that dig deep into

the methane generation zone and periodically ventilate seawater that contains sulfate through their burrows results in cross-stratal fluid conduits (e.g., Forster and Graf, 1992; Ziebiset al., 1996).

Overall, the main factor governing seep carbonate stacking in Aurel thus appears to be the contrast in $m^3$ scale permeability between the PBH and its host sediment (Fig. 15C). Sedimentation of the background facies defines a strongly anisotropic permeability tensor on a $m^3$scale, with low vertical permeability due to the

fine-grained marls and high lateral permeability before lithification occurs due to the presence of granular limestone beds. At the same scale in the seep carbonate column, the *Thalassinoides/Spongeliomorpha* burrow network results in high vertical permeability (e.g., Gingras et al., 2012), whereas lateral permeability is generally lowered by bioturbational blurring of thinner granular limestone beds into the marls. Only granular beds significantly thicker than the mixing depth of near-surface endobenthic organisms remain to a large proportion in

their original state (e.g.,  Wheatcroft, 1990; Wetzel, 2009) and can retain enough permeability to act as drains, leading to self-sealing from top of bed, lateral deflection of incoming fluids and lateral shift of the stacking pattern.

**6. Conclusion**

The study of the Aurel pseudobioherm, a 15-m-thick, 8-m-diameter columnar seep carbonate body encased in a >500 m-thick marl succession, was undertaken in order to define the factors that promoted its vertical growth, apparently contradicting the "self-sealing" paradigm. One key factor appears to be intense bioturbation by decapod crustaceans through the following mechanisms:

• The specific morphology of the deepest tier of burrows includes low-inclination to horizontal segments

that could block passive infill by sediment falling from the seabed or shallower bioturbation tiers, preserving a network of open tubes below.

• The areal density of burrows, ca. 400 burrows/m², was sufficient to ensure cross-cutting between successive generations of burrows, making a connected network thick enough to reach deep below the base of the SMTZ and thereby providing unrestricted vertical fluid communication through the whole carbonate body, in

other terms giving a very high overall vertical permeability to the PBH.

• The host sediment is dominated by fine-grained marls resulting in a very low vertical permeability. In contrast, intervening cm-thick laterally continuous limestone beds make up to a few % of the formation and creating a relatively high lateral permeability. At the location of the PBH, these thin beds were homogenized

through biodeformation in the upper 5-10 cm below seabed, blocking lateral fluid circulation inside the PBH and
into background drains.

- Conversely, the few >10-cm-thick limestone beds were not fully homogenized, leading to self-sealing
and (possibly upslope) lateral PBH migration by a few meters where the PBH crosses the thicker beds.

The critical factors thus appear to be the adequate type and abundance of burrows; these foster
concentration of compaction fluid flow into the PBH with positive feedback on the development of endobenthic
life. Decapod crustacean-promoted focusing of fluid flow into a seepage area could be the antidote to self-sealing
and promote sustained vertical growth of seep carbonate bodies. This type of bioturbation, known to be common
at hydrocarbon seep sites, may thus be a small-scale critical factor fixing a seepage site at the same point for long
spans of time, as has commonly been observed on seismic sections worldwide.

## Acknowledgements

We thank Total Exploration and Production for sponsoring Jean-Philippe Blouet's PhD, for granting
permission to publish this research, and for many insightful discussions. Dr. Eric Gaucher in particular is
thanked for his constructive comments on the SE France 'carbonic province', as well as Dr. Stéphane Teinturier
for preliminary fluid inclusion analysis. The field campaign was made possible by Dr. Philippe Artru, who
guided us to the outcrops where he conducted his PhD work and first discovered the Aurel pseudobioherm back
in the 60's. Philippe also brought us many original ideas on the regional geology of a basin on which he
conducted research for some 10 years of his life. Rolland Oddou, a distinguished collector of septaria
concretions form the Terre Noires formation, generously shared with us his extensive knowledge of the regional
outcrops.  The personnel of the University of Fribourg are thanked for countless advice, in particular Bernard
Grobéty for the interpretation of XRD spectra, and Silvia Spezzaferri for identification of microfossils. Ivano
Aiello and Martin Hovland are thanked for the comments they provided on the PhD thesis chapter version of this
article. Maciej Bojanowski is thanked for insightful discussion during a field trip sponsored by the International
Association of Sedimentology. Thibault Renard and Simon Vuadens are acknowledged for their help in the field
in the frame of their master theses. Thin Section Lab and its personnel (Toul, France) are thanked for providing
access to their facilities.



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

**Figures**

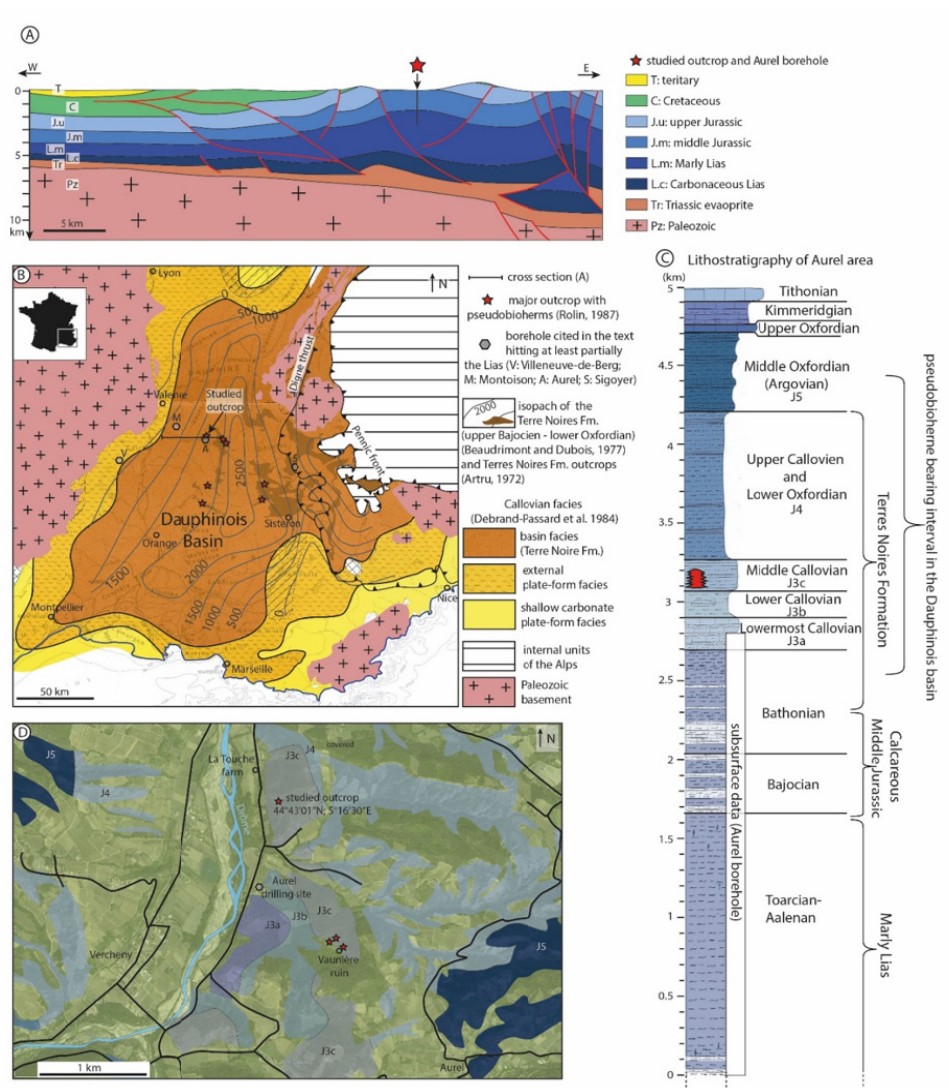

**Figure 1: Geological setting of the studied outcrop. A) Cross-section from the edge to the center of the Dauphinois Basin running across the Aurel borehole (modified from by Wasseron and Bessereau, 1999). B) Map of the sedimentary facies in the Dauphinois basin during the Callovian (Debrand-Passard, 1984). The basin facies, corresponding to the Terres Noires Formation, are restricted to the central part of the basin. The total thickness of the Terres Noires Formation (middle Bathonian up to lower Oxfordian) is indicated (Beaudrimont and Dubois, 1977). The outcrops of the Terres Noires Formation (Artru, 1972) and the major pseudobioherm location are marked by red stars (Rolin, 1987). C) Lithologic log of the Aurel area (Flandrin 1974). The Aurel borehole crossed through the carbonate-dominated Dogger and the marly upper Lias intervals. The pseudobioherms of the Aurel area are located in the middle Callovian interval. The stratigraphic distribution of the pseudobioherms in the Dauphinois Basin, extending from the upper Bathonian to the middle Oxfordian, is indicated. D) Geological map of the Aurel area (modified from Flandrin 1974). The names of the formations are indicated in the lithologic column. Several poorly**







**exposed pseudobioherms are present near the Vaunière ruin; the described pseudobioherm is located near La Touche farm.**



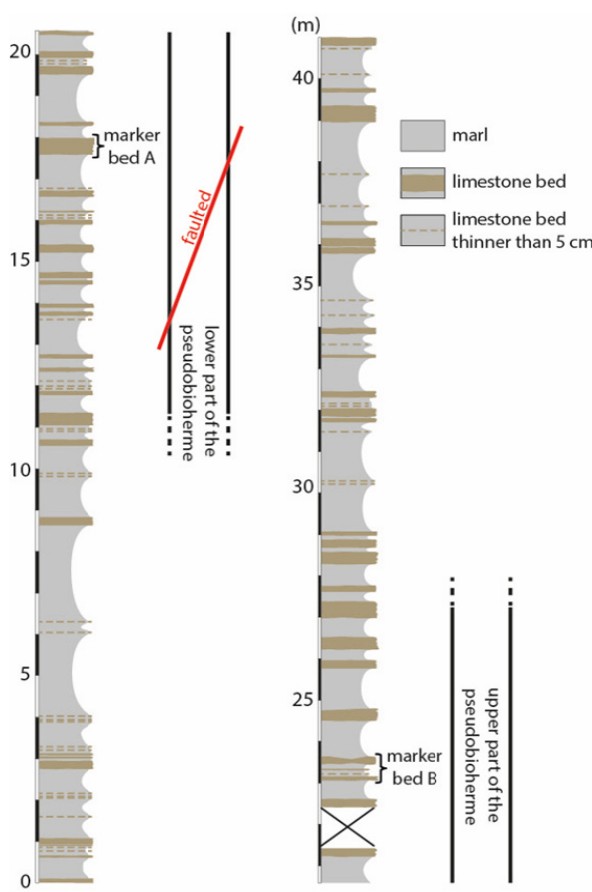

**Figure 2: Lithologic log of the middle Callovian in the pseudobioherm-bearing interval. The logged section is located about 20 m to the west of the upper part of the pseudobioherm, unaffected by faulting. The positions of the protruding marker beds A and B, easily recognizable in the pseudobioherm, are indicated.**




**Figure 3: A) and B), photo and interpretation of the southern flank of the Aurel PBH. Dashed lines represent the interpreted axis of the PBH and highlight its evolution during the growth of the carbonate column. C) and D), photo and interpretation of the eastern flank of the PBH; LST stands for "limestone".**




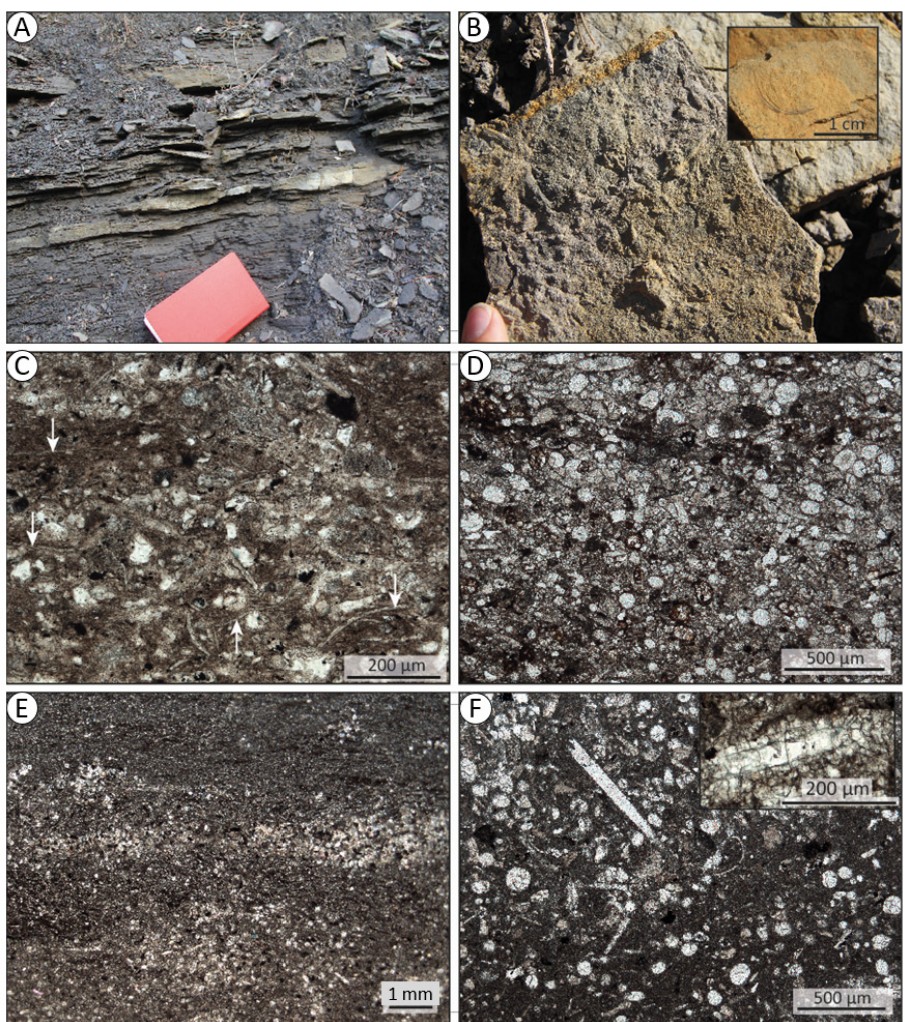

**Figure 4: Background facies F1. A) Typical section of a limestone bed showing the platy fabric. B) Surface of a slab with radiating bioturbations (*Asterosoma*). The inset shows the bivalve *Bositra*, the only relatively common fossil in F1. C) Marl with numerous filamental structures interpreted as disaggregated *Bositra* shells. D) Limestone beds show the same microfossil content as the marl but in a 10 µm microsparitic matrix. E) Laminated structure of the limestone beds produced by an alternation of packstone and wackestone textures. F) Marls showing numerous recrystallized microfossils in a micritic matrix. The circular structures are interpreted as calcispheres and/or radiolarians. The inset shows a tubular structure interpreted as a sponge spicule.**




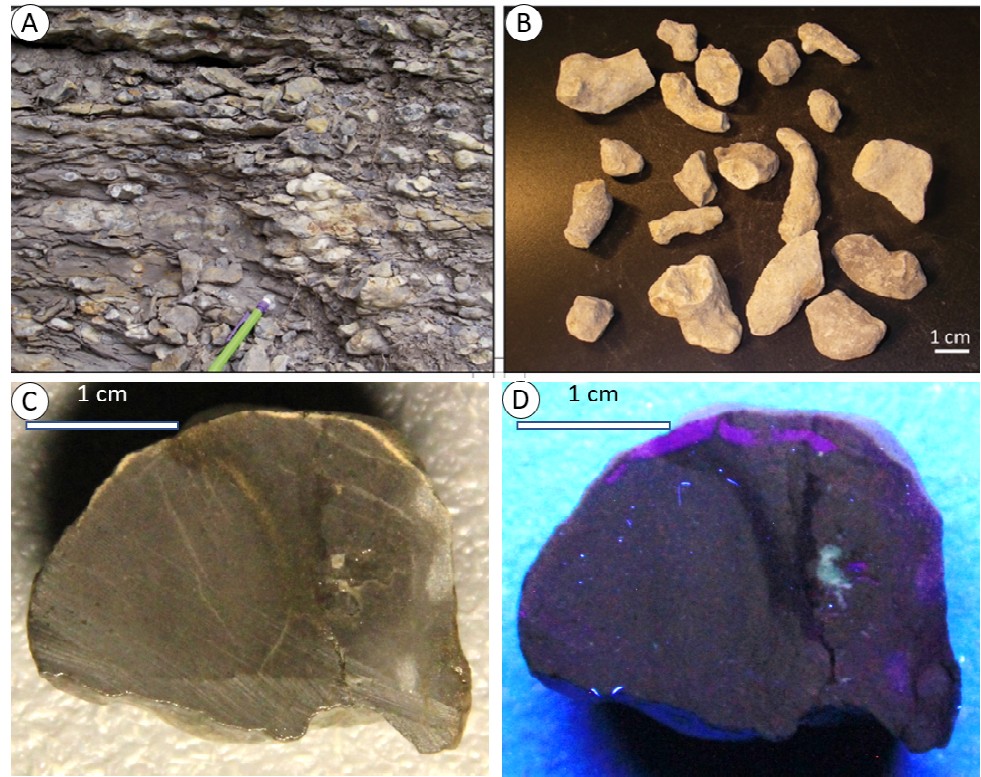

**Figure 5: Facies F2, nodular marl. A) General aspect at the outcrop. B) Individual nodules and nodule clusters. C) and D) cross-section of a nodule under natural and UV light, respectively. The darker (C) non-fluorescent (D) vertical parting in the middle shows that what looks like an individual nodule is actually a cluster of two.**



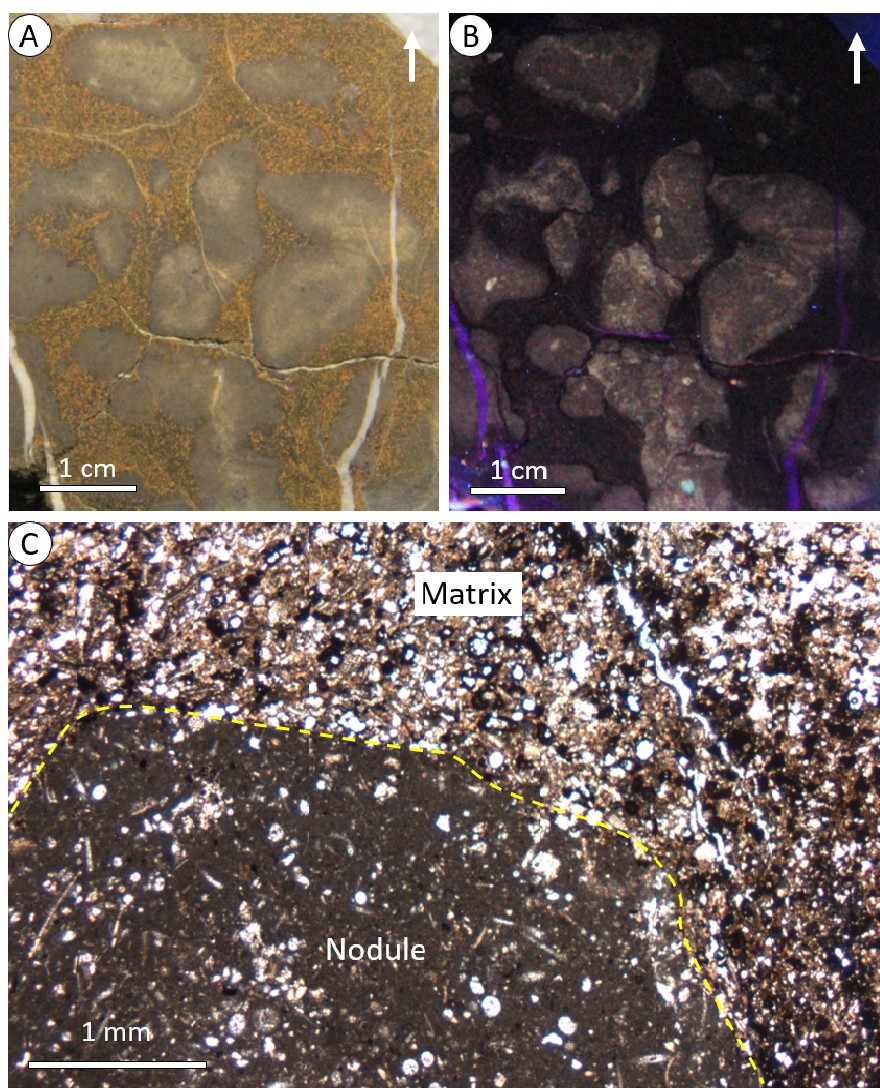

**Figure 6: Facies F3, nodular limestone. A) and B), vertical section of a sub-block of F3. See descriptions in the text. The arrow points to the top of the sample. C) Photomicrograph showing the contact between a nodule and its matrix (highlighted by a yellow dashed line). Grain density seems higher in the matrix than in the nodule, suggesting that the cementation of the nodule predates most of the compaction.**




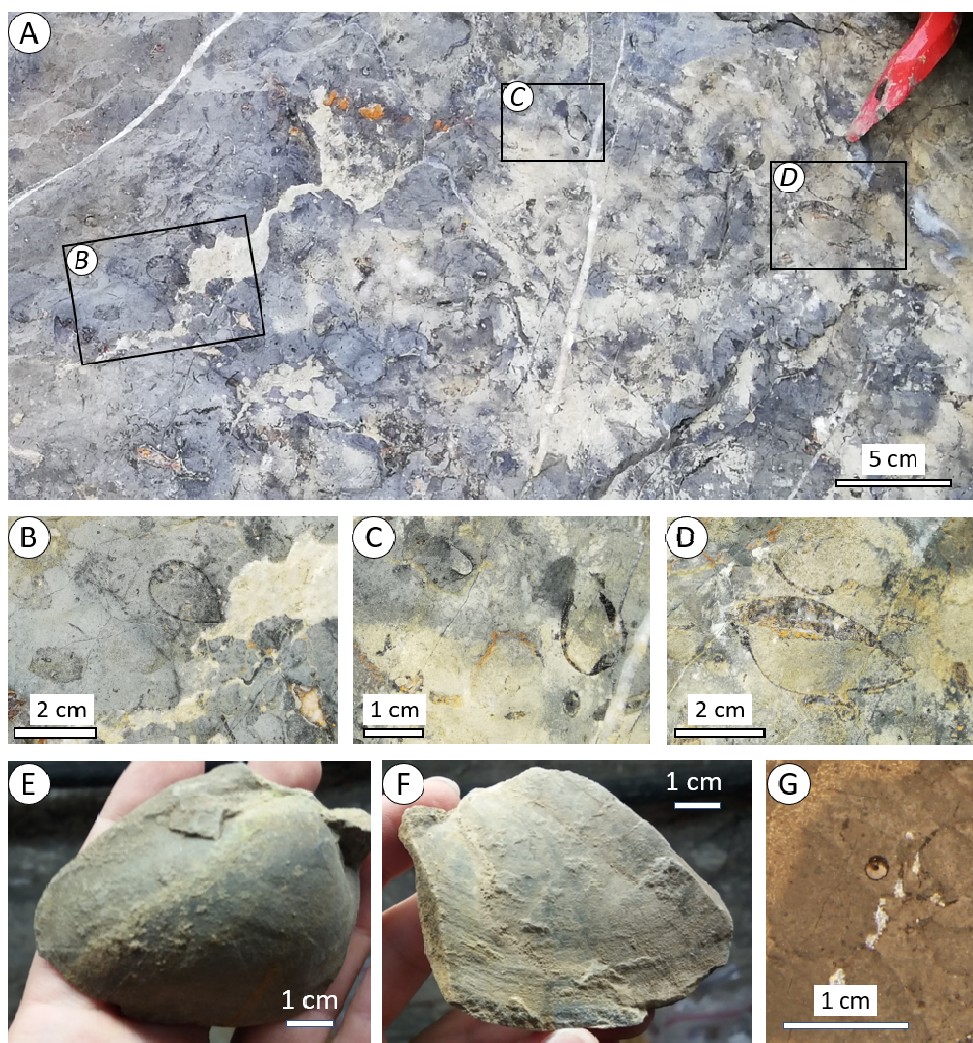

**Figure 7: Macroscopic aspect of Facies F4, massive limestone, highlighting its specific fauna. A) Weathered surface showing three bivalves. B) to D) are close-ups on the three bivalves, apparently belonging to different species. D) shows a bivalve with geopetal filling. E) and F) are the two valves of the only lucinid that could be retrieved from the PBH. G) section of a small gastropod.**


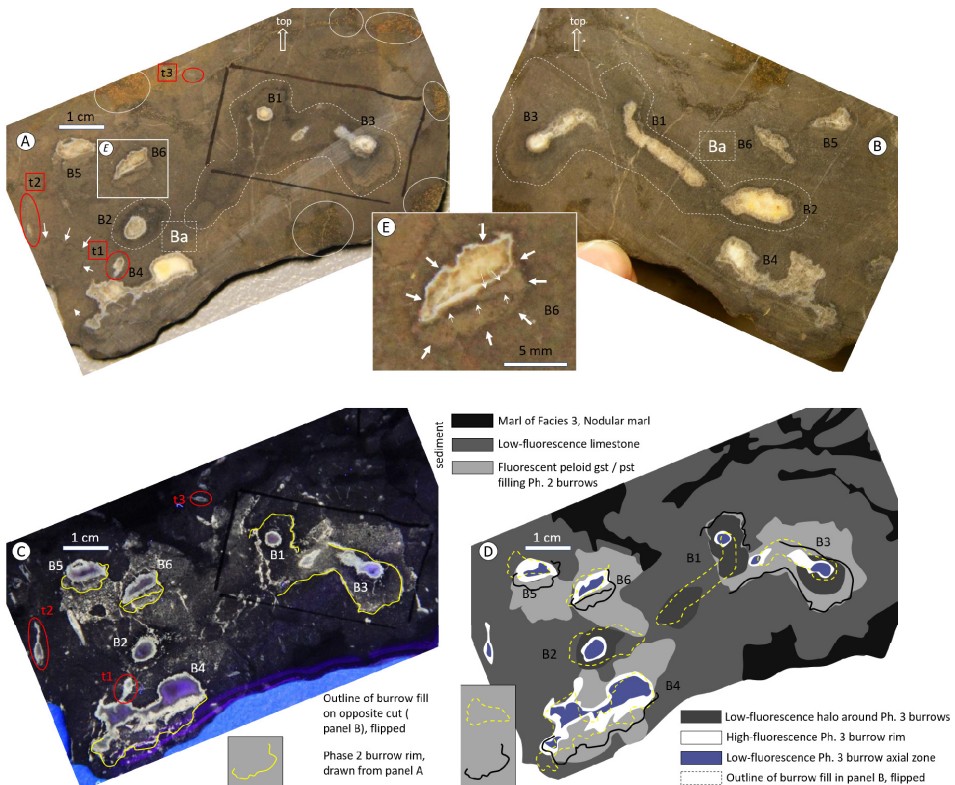

**Figure 8: Burrow architecture in F4. A) and B), opposing sides of a single sawcut, ca. 3 mm apart. Black rectangle drawn for thin section selection. White ellipses on the side indicate patches of marl with rusty speckles, i.e. transition to F3. Three types of burrows are identified: large burrows, one of which is identified as Ba and surrounded by a dashed white line; medium burrows, filled or partly filled with white-yellowish cements and identified as B1-6; and small burrows t1-3 circled in red. White arrows in (A) point at intra-sediment dark linings. C) UV light view of the section shown in (A). Yellow outlines indicate the location of the main dark linings, as traced on (A). D) Line drawing of (C) showing the main burrows. The yellow dotted line represents the outlines of B1-6 in (A) and (C), giving an insight into 3D burrow geometry; gst / pst in the legend stand for grainstone and packstone, respectively. See details in text. E) Close-up view of B6 in (A) showing the burrow surrounded by a dark lining (bold arrows) and half-filled by geopetal sediment split in two and covered by other dark linings (thin arrows).**






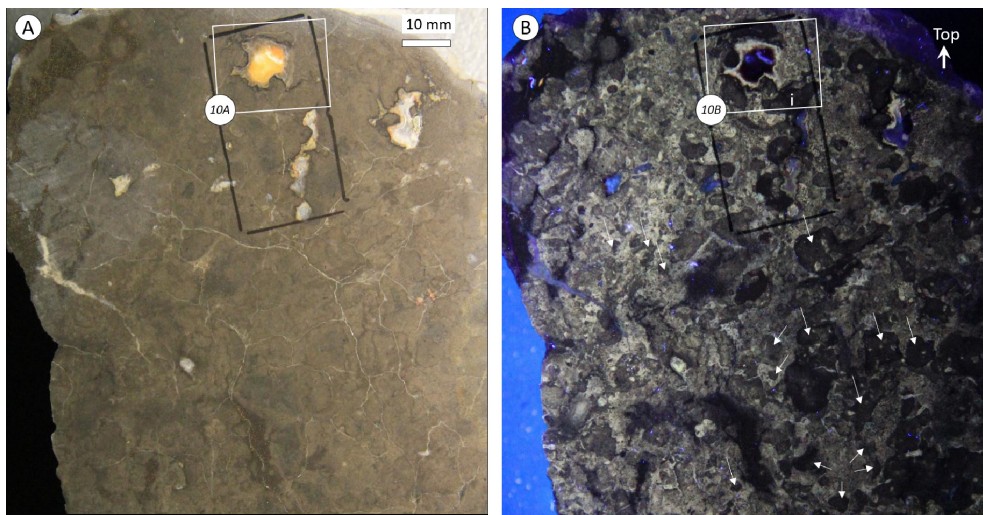

**Figure 9: Sediment texture in heavily bioturbated samples. A) natural light and B) UV fluorescence views. The black rectangles were drawn for positioning the thin section cutting. White arrows in B) indicate micrite intraclasts. Note the variability of the intraclasts in roundness and size.**






**Figure 10: see caption next page**





**Figure 10: A) and B), close-up of Figs 9 A-B showing the details of burrow infill. Orange arrows in (A) indicate the**

**dark lining surrounding the burrow on the left side and expanding on the right side to include peloid grainstone (PG). Yellow arrows indicate a thin, continuous, milky white lining which follows the dark lining in the left part in the image but in the right part follows the sediment-cement contact. B) UV fluorescence view of the same section. Int stands for micrite intraclasts, EC for early cements, LC for late cements. The white rectangle shows the location of C). The intraclasts, like nodules in F2-3 have a low level of fluorescence, in contrast with brightly fluorescent peloid**

**grainstone (the less fluorescent spots inside are individual peloids). The brightly fluorescent rim coincides with the cement ring limited by the dark rim on the outer side and the white rim on the inner side (C), thin section view of the same sample. The dark lining (orange arrows) corresponds to the superposition of brown and clear microsparite layers, brown microspar (BMSpar) directly overlying burrow wall sediment. The white rim (yellow arrows) consists of beige chalcedony, see details in Fig. 11. White arrows point to shell fragments, PW stands for peloid wackestone.**

**Dol 2 refers to late saddle dolomite cement. D) and E) are representative photomicrographs of the peloid grainstone in F4: D) view in plane polarized light, showing the internal texture of the peloids; E) UV epifluorescence view showing that the microspar cement is fluorescent in contrast with low-fluorescence peloids.**




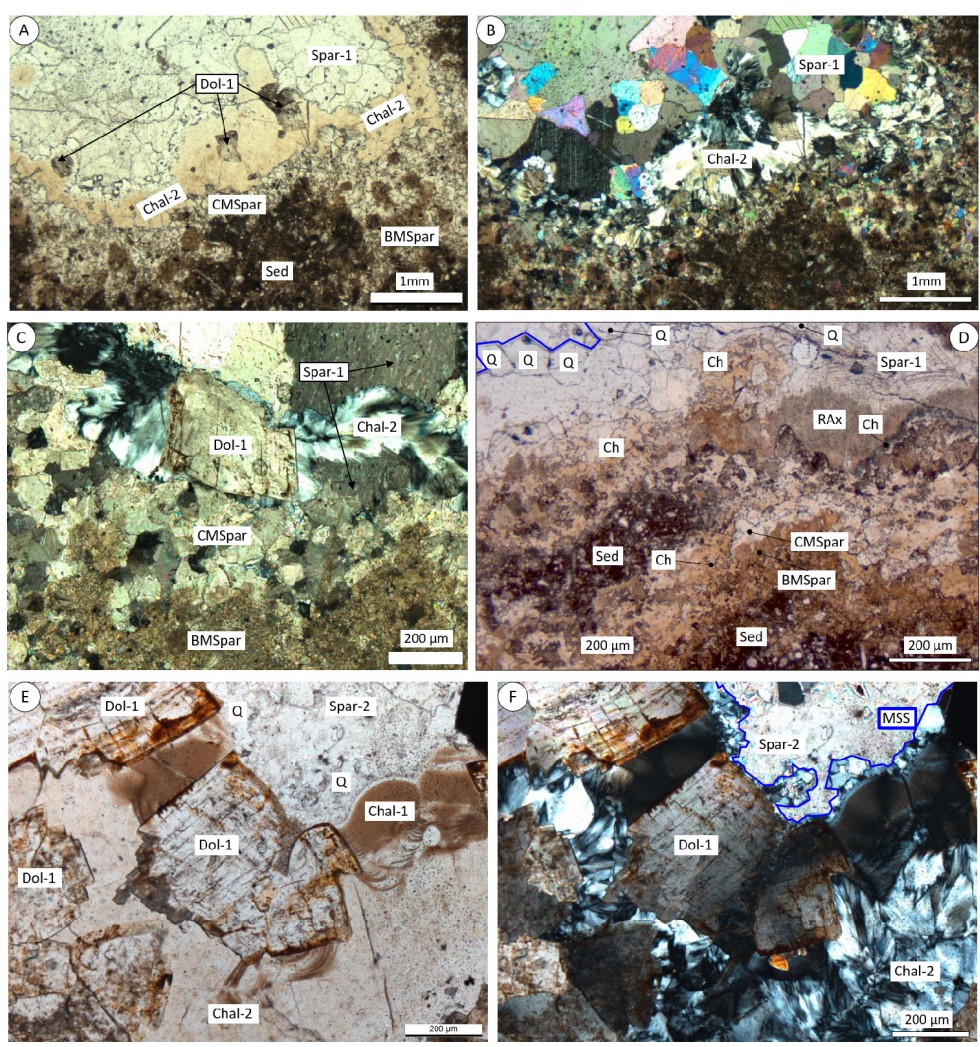

**Figure 11: post-sedimentary cements. A) and B) are respectively plane polarized view and cross polarized view of a phase-3 burrow lumen and its host sediment. Note the continuous rim of beige chalcedony (Chal-2), associated with saddle dolomite crystals (Dol-1). C) Cross-polarized light, higher magnification view to show the relationships between Chal-2 and carbonate cements. In the right part of the image, Chal-2 clearly cuts across a single crystal of sparite (Spar-1), indicating it is a late replacement mineral. D) Plane polarized view of two stacked sediment-cement**

**sequences, showing an alignment of euhedral quartz crystals Q roughly parallel to the burrow wall. It also provides an example of the radiaxial grey calcite crystals (RAx) locally capping the Sed-BMSpar-CMSpar succession; the blue zigzag line follows the upper limit of euhedral quartz crystals. E) and F), are respectively plane polarized view and cross-polarized view of the main phases of silicification. Chal-1 is botryoidal chalcedony that precipitated freely at the tube wall. Q denotes euhedral quartz crystals that also overgrew a free surface, while Chal-2 and the associated saddle**

**dolomite Dol-1 replaced both pre-existing carbonates and Chal-1. Note the extinction pattern of Chal-2, which characterizes "flamboyant chalcedony". The blue zigzag line noted MSS is the main silicification surface. See text for details.**




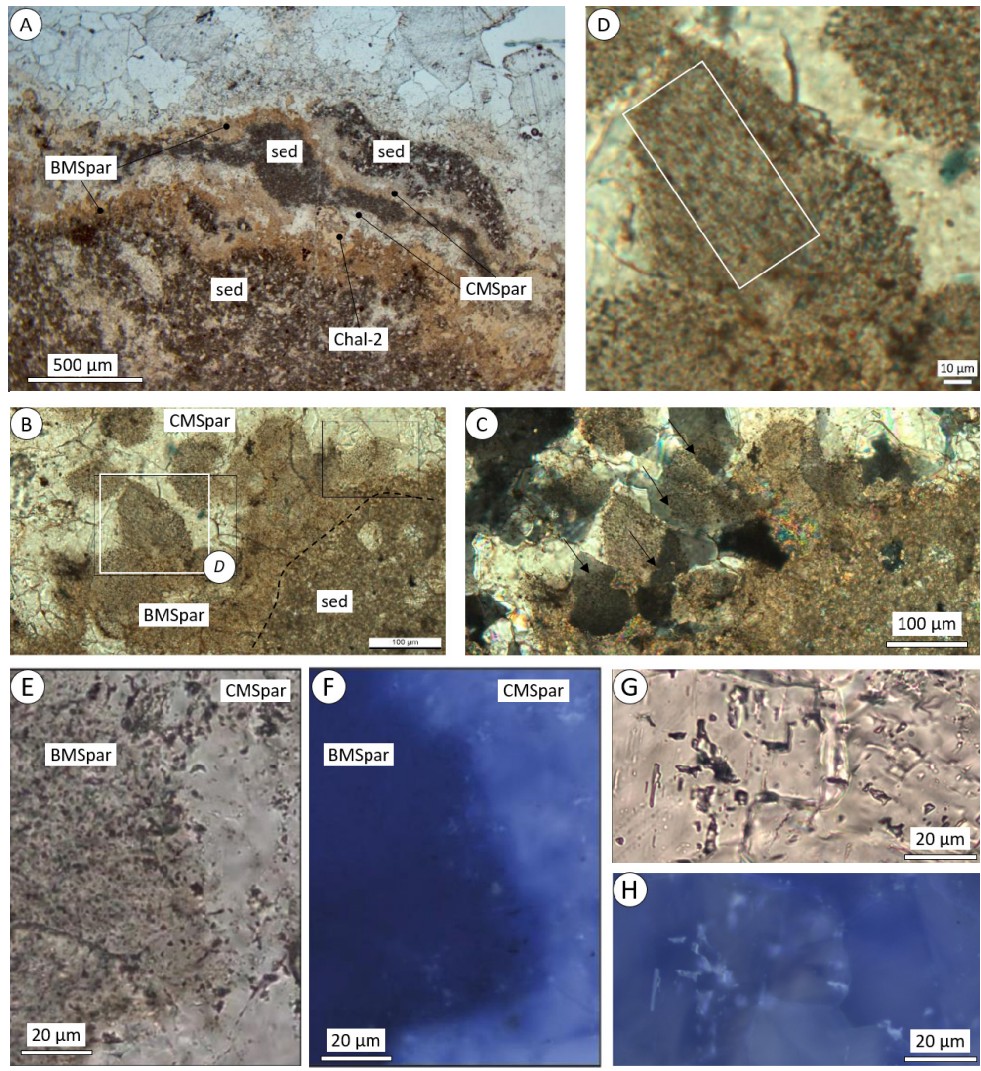


**Figure 12: thin section views showing details of sediment-cement sequences. A) Repeated sequence with sediment (sed), brown microsparite (BMSpar) and clear microsparite (CMSpar), locally overprinted by chalcedony (Chal-2); B) close-up view of the BMSpar / CMSpar contact: BMSpar makes filamentous "bushes". The dashed black line highlights the sediment-BMSpar contact. C) Cross-polarized light view of B) showing that individual microspar**


**crystals straddle across BMSpar/CMSpar boundaries (arrows). D) Close-up on one of the filamental bushes. The white rectangle shows the trend of filaments, which appear as elongate brown/gold inclusions in a clear matrix. E) and F), respectively plane polarized light view and UV epifluorescence views across a BMSpar/CMSpar contact. Fluorescent inclusions near the margin of BMSpar follow the oblique contact between BMSpar and CMSpar inside the thin section. G) and H), plane polarized light and UV epifluorescence views of typical CMSpar, in which**


**fluorescence comes from both fluid inclusions (bright white) and from the microspar itself (dull yellow).**



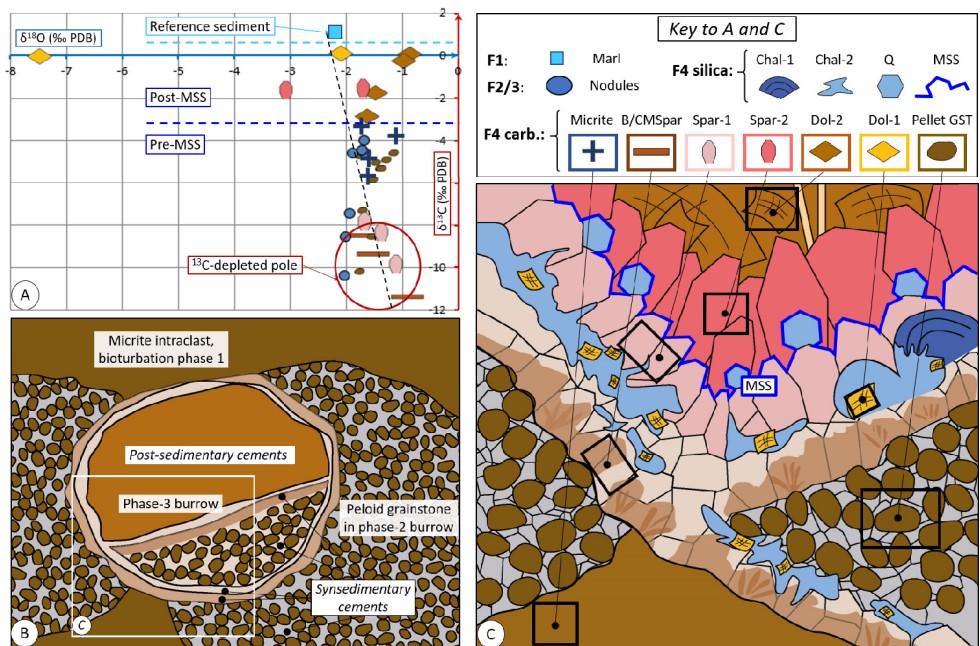

**Figure 13: comparison between carbon / oxygen isotopy and microfacies. A) Delta** $^{13}$**C-**$\delta^{18}$**O plot of the various carbonate phases of the Aurel PBH, sedimentary and diagenetic. B) general texture of F4 with the successive phases of biodeformation / bioturbation. C) Generalized sketch of the geometric relationships between samples selected for isotope analysis in facies 4; the position of samples from facies 1-3 is shown in Fig. 14. Black boxes symbolize the minimum size of samples allowed by the drilling bit used, some of which like peloid grainstone and BMSpar/CMSpar mix several phases. GST for grainstone, MSS for "main silicification surface", see explanations in text.**






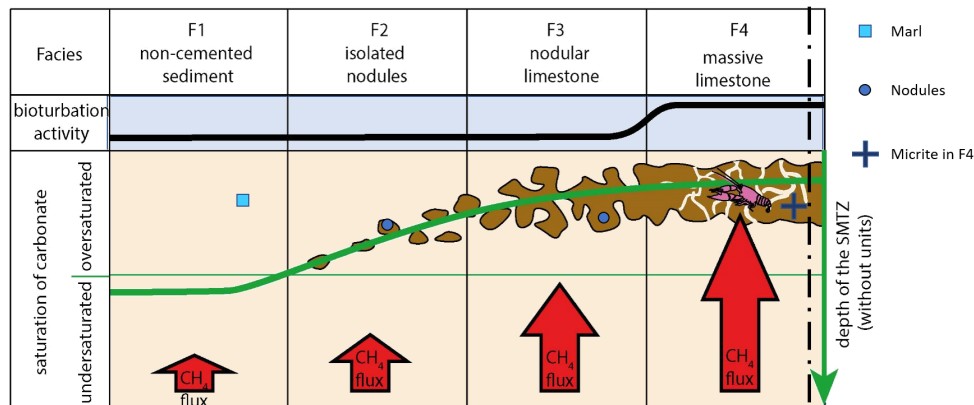

**Figure 14: lateral evolution of the sulfate-methane transition zone (SMTZ) and MDAC precipitation from background sediment towards the PBH. The vertical dashed line indicates the axis of the PBH, symbols refer to the isotope cross-plot in Fig. 13A. Brown shapes are the zones of active MDAC precipitation.**





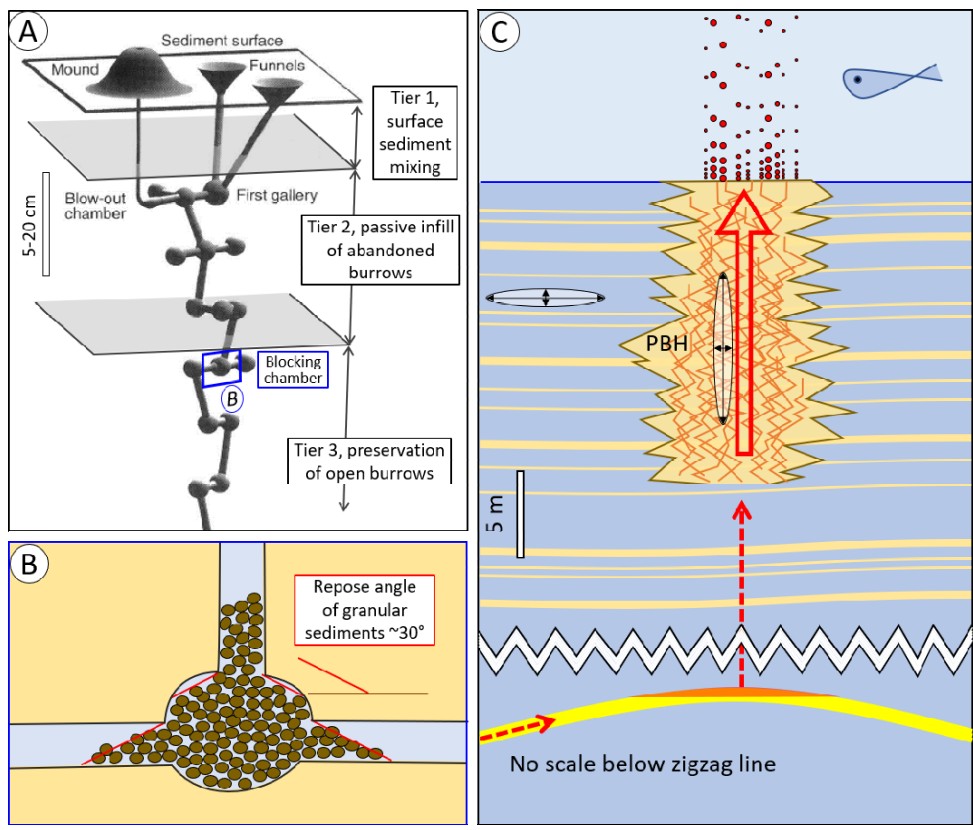


**Figure 15: Fluid flow in and around the PBH. A) modified from Ziebis et al. (1996): typical architecture of a recent decapod crustacean burrow (*Callianassatruncata*) characterized by the presence of numerous low inclination / horizontal segments, in that way comparable to Aurel tier-3 burrows. Grey surfaces indicate the hypothetical limits between bioturbation tiers 1-3 in Aurel. The blue polygon shows the shallowest chamber in which any granular**

**sediment shed down from tier 2 will be blocked. B) Close-up of the blocking chamber: once lateral spillout from the chamber reaches the repose angle, any additional incoming granular material from above will continue piling up in the shaft, while the deeper parts of the burrow are preserved open. C) Sketch of fluid circulation in and around the Aurel PBH during its growth. Lower part, without scale: hypothetical deep structure focusing late thermogenic methane into the PBH; upper part, general gas circulation pattern inside the PBH. Orange zigzag lines represent**

**decapod crustacean burrows making a connected network down to the bottom of the PBH. The white ellipses represent the permeability tensor in the PBH and in its host sediment, with the principal axes of minimum and maximum permeability shown by black arrows. Red dashed arrows at depth indicate hypothetical gas migration pathways from a source rock into the structure and u from the structure to the base of the PBH; the red arow within the PBH symbolize the overall vertical flow patter of the gas, likely with episodic escape of excess gas at the seabed**

**(red bubbles in the water column).**





**Highlights**

- We describe facies architecture of a 15m thick and 10m wide columnar carbonate chimney

- The column is characterized by a dense network of subvertical crustacean burrows


- An estimated 2-500 burrows/m² remained open until long after burial of the column

- Intra-cement burrows are depleted in 13C, indicating AOM in the shallow subseabed

- The open burrow network promoted vertical growth of the methane seep carbonate chimney



**Appendix 1**

| Sample # | $\delta^{13}C$ (‰PDB) | $\delta^{18}O$ (‰SMOW) | $\delta^{18}O$ (‰PDB) | Description |
|---|---|---|---|---|
| 80 | 1.04 | 28.64 | -2.20 | marl |
| 2_1 | -8.55 | 28.84 | -2.00 | micrite |
| 2_2 | -10.16 | 29.09 | -1.76 | peloidal grainstone |
| 2_3 | -9.38 | 29.34 | -1.52 | microsparite as rim of cavity |
| 2_4 | -7.27 | 29.15 | -1.70 | peloidal grainstone |
| 2_5 | -8.51 | 29.31 | -1.55 | peloidal grainstone |
| 2_7 | -7.83 | 29.19 | -1.67 | Sparite 1 |
| 2_8 | 0.05 | 30.03 | -0.85 | Dolomite 2 |
| 2_9 | -7.49 | 28.91 | -1.94 | micrite |
| 3_1 | -3.30 | 29.12 | -1.73 | micrite |
| 3_2 | -5.84 | 29.34 | -1.52 | peloidal grainstone |
| 3_3 | -4.87 | 29.58 | -1.29 | peloidal grainstone |
| 3_4 | -5.69 | 29.24 | -1.62 | micrite |
| 3_5 | -5.26 | 29.47 | -1.39 | peloidal grainstone |
| 3_7 | -0.06 | 23.20 | -7.48 | Dolomite 1 |
| 3'_1 | -4.57 | 29.73 | -1.14 | peloidal grainstone |
| 3'_4 | -1.65 | 27.72 | -3.09 | Sparite 2 |
| 15_1 | -4.83 | 29.26 | -1.60 | micrite |
| 15_2 | -5.00 | 29.39 | -1.47 | peloidal grainstone |
| 20_1 | -5.28 | 29.50 | -1.36 | peloidal grainstone |
| 2,1_1 | -8.53 | 29.21 | -1.65 | microsparite as rim of cavity |
| 2,1_4 | -8.34 | 29.50 | -1.36 | Sparite 1 |
| 2,1_5 | -9.83 | 29.77 | -1.10 | Sparite 1 |
| 2,1_6 | 0.05 | 28.76 | -2.08 | Dolomite 1 |
| 2,1_7 | -0.27 | 29.91 | -0.97 | Dolomite 2 |
| 2,2_1 | -11.39 | 29.98 | -0.90 | microsparite as rim of cavity |
| 2,2_2 | -1.55 | 29.15 | -1.70 | Sparite 2 |
| 2,2_3 | -1.78 | 29.39 | -1.47 | Dolomite 2 |
| 3,1_1 | -2.85 | 29.24 | -1.62 | Dolomite 2 |
| 3,1_2 | -3.77 | 29.77 | -1.10 | micrite |
| 3,1_3 | -4.67 | 29.09 | -1.76 | peloidal grainstone |
| 1_1 | -2.42 | 28.98 | -1.87 | micrite |
| 84_1 | -4.61 | 29.20 | -1.66 | micrite |
| 84_2 | -4.61 | 29.15 | -1.70 | micrite |
| 84_3 | -4.47 | 29.18 | -1.67 | micrite |
| 84_4 | -3.99 | 28.81 | -2.03 | micrite |
| 84_6 | -10.42 | 28.94 | -1.91 | micrite |
