# Peer review of "What makes seep carbonates ignore self-sealing and grow vertically? The role of burrowing decapod crustaceans"

_Solid Earth, 2020_

## Referee Comment (RC1) · Anonymous Referee #1 · 1 Dec 2020

General comments I have carefully read this manuscript and found it to be of interest. The manuscript focuses on the role of bioturbation in creating fluid pathways at methane seeps. Burrows, in particular those of decapod crustaceans, are suggested to favor the vertical aggregation of seep deposits despite of the self-sealing effect of carbonate crust formation. By providing a detailed description of the burrow network of a Jurassic seep deposit, this works adds to the literature on seep environments. Two of my comments are of a more general nature.

(1) It is a missed chance that silicification, formation of chalcedony, and precipitation of euhedral quartz crystals are not put into perspective with the same phenomena

other seeps described in the literature. Like for the studied Aurel seep deposit, silicification and silica authigenesis have been observed to postdate the precipitation of methane-derived carbonate, but to predate later diagenetic phases lacking C-13 depletion. This relationship has now been described for many seep deposits and a hypothesis to explain the observed paragenetic sequence has been developed (Kuechler et al., 2012, Lethaia 45, 259-273; Smrzka et al. 2015, Palaeogeography Palaeoclimatology Palaeoecology 420, 13-26). Discussing the context of silicification and silica authigenesis will help to elaborate the postulated timeline of events.

(2) It is mentioned that carbonate crusts may grow downward at seeps (Bayon et al. 2009). Yet possible implications of downward aggregation are not discussed. If, indeed, seep deposits will preferentially grow downward into the sedimentary column, the impact of bioturbation on maintaining fluid flow on longer time scales will be more limited than suggested in this manuscript. Based on the study of mesofabrics of seep limestones, it had been suggested that the aggregation of methane-derived carbonate may proceed downward (Greinert et al. 2002, Int. J. Earth Sci. 91, 698-711; Peckmann et al. 2002, Sedimentology 49, 855-873). While downward growth may indeed occur, the work of Liebetrau et al. (2014, Int. J. Earth Sci. 103, 1845-1872) suggested that upward growth is typically more pronounced. These findings, particularly the work of Liebetrau and co-workers, should be discussed and their relevance should be put into perspective to the inferred role of bioturbation in the formation of the Aurel seep deposit. The authors seem to suggest preferential upward aggregation in case of the Aurel deposit, but this needs to be clarified and should be discussed in more depth.

I found it difficult to follow the captions of the figures with photomicrographs. Figure captions should be self-explanatory on the one end – these are not – and should be succinct on the other end – which they are not either. Consider to focus on what is really needed for the description of the micrographs and what can be moved to the main text. The paragenetic sequence should be apparent from the caption itself.

The authors manage to get the message across, but the standard of the English is less

than ideal. The manuscript would benefit from linguistic editing of a native speaker.

In conclusion, I recommend publication of this interesting work after moderate to major revision.

A brand new publication that should be considered during revision: Gay et al. (2020) Poly-phased fluid flow in the giant fossil pockmark of Beauvoisin, SE basin of France. BSGF-Earth Sciences Bulletin 2020, 191, 35.

Specific comments on the manuscript Note: I do not use special characters in this web-based review (1) Line 45: I do not want to be nit-picking, but the precipitation of dolomite requires magnesium ions in addition to calcium. (2) Line 141: Chemosymbiosis can only be assumed in case of ancient taxa. (3) Lines 239 to 240: What would be an "altered peloid" – please specify. (4) Line 264: The work of Rolin et al. (1990) is not the latest publication on the Beauvoisin lucinids. A new species has been formally described by Kiel et al. (2010; Zootaxa 2390, 26-48). (5) Result chapter, petrography (e.g. page 10): The circumstance that the mineral phases of the paragenetic sequence are not described in chronological sequence impedes comprehensibility. (6) Chapter 5.2: The reasoning about carbon stable isotopes is mostly okay. Yet, based on the carbon stable isotope data alone, a relation to methane seepage cannot be proven. The described limestone deposit should be compared with the nearby Beauvoisin seep deposits, for which the involvement of anaerobic oxidation of methane in carbonate formation has been proven with lipid biomarkers. (7) Lines 509 to 510: "limestone column" – The sedimentary strata do not consist of limestone only. (8) Line 533, and throughout: "MDAC" – This abbreviation has not been introduced. But why would you like to use it anyway? 'Seep carbonates' are one type of 'methane-derived authigenic carbonates'. Carbon-13 depleted phases of septarian concretions are another example. The designation 'seep carbonate' is consequently more specific than the acronym 'MDAC'. (9) Line 535: Silicification predates the formation of burial cement. I would not call such silicification 'late', although it is admittedly later than the formation of methane-derived cement. (10) Line 537: "calcite precipitation" – You cannot exclude that much of the

calcium carbonate precipitated as aragonite cement like at most modern and Phanerozoic seeps. (11) Line 543: "brown color of BM spar – Based on its position in the paragenetic sequence, I consider it more likely that this phase corresponds to primary yellow or brownish aragonite (e.g., Zwicker et al. 2015; Marine and Petroleum Geology 66, 616-630). (12) Lines 554 to 572: This is where authigenic silica formation and silicification at seeps needs to be discussed (see general comment). (13) Chapter 5.3.3.: This chapter does not add much to the manuscript – the discussion is vague to say the least. (14) Line 607: "burrowers feeding on chemosymbiotic microbial communities" – 'Chemosymbiosis' refers to the association of a metazoan host (e.g., bivalve, tubeworm) with endosymbiotic, chemotrophic bacteria. The term 'chemosynthetic' would work in this instance. (15) Lines 609 to 610: See also Zwicker et al. (2015, see above) for the role of burrows as part of the shallow plumbing systems in sediments affected by seepage. (16) Line 640: "methane generation zone" – This should be replaced by 'methanic sediments' (i.e., sediment containing methane). Methanogenesis (i.e., methane formation) occurs at greater depth, although minor methanogenesis may also occur at or close to the sulfate-methane transition zone. (17) Fig. 7 (E) and (F): Could this be Beauvoisina carinata (see comment 4)? The shell seems pretty asymmetric, maybe more asymmetric than in B. carinata.

Technical corrections and suggestions (1) Line 26 and throughout: omit blank between numbers and per mil symbol. (2) Line 30: "post-dating the burial" is an ambiguous formulation. It could be misinterpreted in the sense that this phase formed after uplift during telogenesis. (3) Line 31: "late final blocking" – I do not understand. Do you mean that the fluid pathways have been plugged at some point? (4) Line 45 and 46, and throughout: "H2S and HS" – Why do you use formula instead of words? Before you used the word methane and not its formula. Be consistent. If chemical formula are used, charges (HS-) need to be indicated too. (5) Line 65: improve wording (6) Line 94: 'implies' instead of 'imply' (7) Line 141: "PBH's (pseudobioherms) – You use many abbreviations and acronyms; this does not make reading any easier. What is the benefit of replacing the word 'pseudobioherms' by the abbreviation 'BHPs'? Saving

space? Consider to refrain from introducing yet another new abbreviation. (8) Line 144: Consider to use 'perimeter' instead of 'circle'. (9) Line 171: Same as for per mil. Omit blank between numbers and per cent symbol. (10) Line 246: Add blank before "As". (11) Line 257, and throughout: It is 'gray' in American English. (12) Line 261: add blank after "of" (13) Line 268: "fabric is " not "fabricis" (14) Line 294: "burrows have" not "burrow shave" (15) Line 296: add blank after "Burrow" (16) Line 310: add blank after "10" (17) Line 313: rather "Taken together" (18) Line 383, and throughout: "main silicification surface (MSS)" – the use of such abbreviation impedes comprehensibility (19) Line 410: "measurements" not "mearurement" when "are" is used (20) Line 412: blank after per mil symbol (21) Line 412: "pole" – I do not understand. Do you mean 'pool'? But even than such wording would be less than ideal. (22) Line 414: blank after "and" (23) Line 422: insert blank after "limited" (24) Line 467: "whereas" instead of "where as" (25) Line 467, and throughout: "depleted values" – Colloquial wording. What is it, a value would be depleted in? A values is a number; in this case 'low values' or 'negative values' would be appropriate. A mineral phase, for example, can be depleted in one isotope, in this case C-13, but not a value. (26) Line 486: delete "depletion" (27) Line 492: add blank before "of" (28) Line 501: insert blank before "signature" (29) Line 531: What is "biodeformation"? Is this a good term? (30) Line 599 to 600: ". . . starting from the top shallow within the seafloor" – improve wording (31) Fig. 10 (B) seems out of focus. (32) Line 1041: Add full stop after 'side'. (33) Fig. 12 (D) seems out of focus. (34) Line 1093: 'gray' in American English (35) Line 1108: add blank between numbers and units (36) Line 1111: '13' in superscript

comments are also provided in a pdf document I will upload

Please also note the supplement to this comment:
https://se.copernicus.org/preprints/se-2020-187/se-2020-187-RC1-supplement.pdf

---

## Referee Comment (RC2) · Anonymous Referee #2 · 7 Feb 2021

General comments: The manuscript describes a Jurassic seep carbonate body cropping out in the Aurel area (SE France basin) and focuses particularly on the control exerted by the bioturbation on the vertical growth of the carbonate body. Authors interpret that intense burrowing by callianassid-type shrimps in the central part of the seep enhanced vertical permeability during a long time, which avoided the self-sealing process in the seep deposits and allowed the vertical aggradation of the carbonate body. This work contributes to a better understand on the sedimentation in seep environments and particularly on the formation of high aggrading carbonate bodies. I find this work interesting and it adds to the knowledge about seep-related processes and products. Therefore, I recommend its publication after moderate to major revision. In

the manuscript, I find particularly well described, interpreted and discussed the sedimentary facies architecture and C isotopes. Nevertheless, I have two major general comments about the origin of the tubular structures and burrowing. 1. Origin of tubular structures: It is presented the tubular structures within the seep carbonates as biogenic, e.g. trace fossils, but a discussion about other possible origins (abiogenic gas conduits) is missing. In this regard, abiogenic conduits have been well documented in the literature, both in present-day and ancient seeps, and some of their complex networks resemble that of the Aurel pseudobioherm. In addition, I find the interpretation as burrows should be supported on more data or evidences (the only macroscopic sample presented correspond to a single 11 cm-long rock fragment) and then a discussion on the origin would be relevant. 2. Burrowing: Burrows are classified in three size categories (large, medium and small) each of which is interpreted (based on cross-cutting relationships) as formed in three consecutive phases at progressively deeper tiers. However, manuscript's Figure 8 shows that large burrow (Ba) contains centered medium burrows (B1-B3) and they present a parallel and no cross-cutting relationship. It seems, at least from that figure, that medium burrows are actually cement-filled holes of the large burrow and not different burrows. Nor do the figures show a clear cross-cutting relationship between small and larger burrows. I think that this is a key point to interpret the temporal and spatial (depth) distribution of the burrows and, therefore, cross-cutting relationships among burrows should be better illustrated or with more figures (they could be in the "supplementary material").

Specific comments: Lines 201: Show the three units in Figure 3. Line 280. It is not clear in Fig. 8 that medium burrows cut through large burrows. Line 289. Indicate figure (Fig. 8?). t3 in Figure 8 is too small to observe concentric bioclast orientation. Lines 313-315: Add reference. Line 324. Smooth wall character does not indicate that it be Trypanites but other criteria as cut bioclasts, etc. Line 335-336. Why does homogeneous micrite-rich fabric reflects high bioturbation if there is no evidence of burrowing? Why sediment homogenization or mixing could not be due to other process, as for example gas bubble ascending? Line 354: Description of microfacies and diagenesis

(section 4.5) is organized in tiers 1, 2, and 3, but these tiers are interpretive, and interpretations should be located in the discussion. Therefore, I recommend to delete them from that section. Moreover, description of carbonate phases would be more understandable if they will be presented following cement stratigraphy. In general, I miss comparison with and references to other papers on seep carbonates and particularly on paragenetic sequences. Line 356: What was bioturbated, the original marl or the later micrite carbonate? Line 397. It is used in this line and through the text "synsedimentary cements" (also "sedimentary cements") to indicate early diagenetic cements. I would be better to use always "early diagenetic cements" in contrast to "late diagenetic cements". Line 495: Most D18O values (table in Appendix I and Fig. 13A) corresponding to saddle dolomites are about -1 to -2 per mil. These values are very strange, are higher than reference sediments and early diagenetic cements, and they are not compatible with hot fluids (>60-80°C) from which saddle dolomites precipitate. Common D18O values for saddle dolomites documented in the literature are around -6 per mil or lower. It needs some discussion. Line 575: "sediment-cement alternations" change by "sediment-cement sequences". Always use the same terminology for the same things. Line 593: Add reference to Fig.3 ("...A and B; Fig.3"). Line 597: It is mentioned the downward growth of concretionary crusts. However, it is not clear whether this interpretation corresponds only to layers A and B or to the entire pseudobioherm. It should be state more explicitly. Line 605: It is mentioned that the axis of vertically stacked carbonates shows two lateral shifts coinciding with marker beds A and B. Then, these shifts are interpreted that hydrocarbon-charged fluids migrated upslope. However, Fig. 3B suggests that the axis of the PBH migrated in opposite directions, first westward and then eastward. How can this apparent contradiction be explained? What was the regional and/or local paleoslope orientation at Middle Callovian times? Line 608: Change "chemosymbiotic microbial communities" by "chemosynthetic microbial communities". Figure 8: The legend of this figure is fragmentary and very complex with a lot of symbols, colors (not easily identifiable), etc. It should be a single and simpler legend.
Technical corrections: - Omit blank spaces front and back "/" and "-"symbols. Revise throughout the document. - Omit blank spaces between number and °C symbol. Revise throughout the document. - Omit blank space between number and per mil and per cent symbols. Revise throughout the document. - Insert blank space between two words. In many places of the text, blank spaces between words are missing. Revise throughout the document.

---

## Author Comment (AC1) · 13 Apr 2021

**Answers to comments of Referee 1**

**Referee 1:** General comments I have carefully read this manuscript and found it to be of interest. The manuscript focuses on the role of bioturbation in creating fluid pathways at methane seeps. Burrows, in particular those of decapod crustaceans, are suggested to favor the vertical aggregation of seep deposits despite of the self-sealing effect of carbonate crust formation. By providing a detailed description

of the burrow network of a Jurassic seep deposit, this works adds to the literature on seep environments. Two of my comments are of a more general nature.

(1) It is a missed chance that silicification, formation of chalcedony, and precipitation of euhedral quartz crystals are not put into perspective with the same phenomena [at] other seeps described in the literature. Like for the studied Aurel seep deposit, silicification and silica authigenesis have been observed to postdate the precipitation of methane-derived carbonate, but to predate later diagenetic phases lacking C-13 depletion. This relationship has now been described for many seep deposits and a hypothesis to explain the observed paragenetic sequence has been developed (Kuechler et al., 2012, Lethaia 45, 259-273; Smrzka et al. 2015, Palaeogeography Palaeoclimatology Palaeoecology 420, 13-26). Discussing the context of silicification and silica authigenesis will help to elaborate the postulated timeline of events.

**Authors:** Thanks for providing this insight into a subject we were not that familiar with, along the key references. The hypothesis proposed by the authors would apply quite well in our case, where siliceous microfossils are abundant in background sediments.

**Referee 1:** (2) It is mentioned that carbonate crusts may grow downward at seeps (Bayon et al. 2009). Yet possible implications of downward aggregation are not discussed. If, indeed, seep deposits will preferentially grow downward into the sedimentary column, the impact of bioturbation on maintaining fluid flow on longer time scales will be more limited than suggested in this manuscript. Based on the study of mesofabrics of seep limestones, it had been suggested that the aggregation of methane-derived carbonate may proceed downward (Greinert et al. 2002, Int. J. Earth Sci. 91, 698-711; Peckmann et al. 2002, Sedimentology 49, 855-873). While downward growth may indeed occur, the work of Liebetrau et al. (2014, Int. J. Earth Sci. 103, 1845-1872) suggested that upward growth is typically more pronounced. These findings, particularly the work of Liebetrau and co-workers, should be discussed and their relevance should be put into perspective to the inferred role of bioturbation in the formation of the Aurel seep deposit. The authors seem to suggest preferential upward aggregation in case of the Aurel deposit, but this needs to be clarified and

should be discussed in more depth.

**Authors:** Our interpretation was probably poorly explained. The hypothesis we made, which we will of course revisit in the light of the references provided and the reviewers' comments, was that the density of callianassid burrows, along with their capacity to block sediment infill at turning chambers (our fig. 17, 17c in particular) meant that they could ensure a connection inside the burrows from the seabed down below the base of the SMTZ. Downward growth would then occur only in the few intervals of isotropic permeability, i.e. limestone beds thick enough not to be fully homogenized by near-surface biohomogenization We will clarify and discuss in ore depth, as requested.

**Referee 1:** I found it difficult to follow the captions of the figures with photomicrographs. Figure captions should be self-explanatory on the one end – these are not – and should be succinct on the other end – which they are not either. Consider to focus on what is really needed for the description of the micrographs and what can be moved to the main text. The paragenetic sequence should be apparent from the caption itself.

**Authors:** OK, thanks for this comment, we will do our best to simplify and focus.

**Referee 1:** The authors manage to get the message across, but the standard of the English is less than ideal. The manuscript would benefit from linguistic editing of a native speaker.

**Authors:** All right, we will seek for appropriate assistance.

**Referee 1:** In conclusion, I recommend publication of this interesting work after moderate to major revision.

**Referee 1:** A brand new publication that should be considered during revision: Gay et al. (2020) Poly-phased fluid flow in the giant fossil pockmark of Beauvoisin, SE basin of France. BSGF-Earth Sciences Bulletin 2020, 191, 35.

**Authors:** OK

**Referee 1:** Specific comments on the manuscript Note: I do not use special characters in this web-based review (1) Line 45: I do not want to be nit-picking, but the precipitation of dolomite requires magnesium ions in addition to calcium.

**Authors:** OK

**Referee 1:** (2) Line 141: Chemosymbiosis can only be assumed in case of ancient taxa

**Authors:** OK

**Referee 1:** (3) Lines 239 to 240: What would be an "altered peloid" – please specify.

**Authors:** Altered by diagnesis / recrystallization, will be specified.

**Referee 1:** (4) Line 264: The work of Rolin et al. (1990) is not the latest publication on the Beauvoisin lucinids. A new species has been formally described by Kiel et al. (2010; Zootaxa 2390, 26-48).

**Authors:** Thank you

**Referee 1:** (5) Result chapter, petrography (e.g. page 10): The circumstance that the mineral phases of the paragenetic sequence are not described in chronological sequence impedes comprehensibility.

**Authors:** The problem we had is that sedimentation, bioturbation and diagenesis are intricately mixed, especially with intra-burrow collapse sedimentation taking place several meters below regional seabed and alternating there with cement growth. We will try to find the most suitable structure to pass the message in the clearest possible way.

**Referee 1:** (6) Chapter 5.2: The reasoning about carbon stable isotopes is mostly okay. Yet, based on the carbon stable isotope data alone, a relation to methane seepage cannot be proven.

**Authors:** fair enough!

**Referee 1:** The described limestone deposit should be compared with the nearby Beauvoisin seep deposits, for which the involvement of anaerobic oxidation of methane in carbonate formation has been proven with lipid biomarkers.

**Authors:** OK

**Referee 1:** (7) Lines 509 to 510: "limestone column" – The sedimentary strata do not consist of limestone only.

**Authors:** OK

**Referee 1:** (8) Line 533, and throughout: "MDAC" – This abbreviation has not been

introduced. But why would you like to use it anyway? 'Seep carbonates' are one type of 'methane-derived authigenic carbonates'. Carbon-13 depleted phases of septarian concretions are another example. The designation 'seep carbonate' is consequently more specific than the acronym 'MDAC'.

**Authors:** Being not a big fan of acronyms, I can only be happy with this comment!

**Referee 1:** (9) Line 535: Silicification predates the formation of burial cement. I would not call such silicification 'late', although it is admittedly later than the formation of methane-derived cement.

**Authors:** OK

**Referee 1:** (10) Line 537: "calcite precipitation" – You cannot exclude that much of the calcium carbonate precipitated as aragonite cement like at most modern and Phanerozoic seeps.

**Authors:** OK

**Referee 1:** (11) Line 543: "brown color of BM spar – Based on its position in the paragenetic sequence, I consider it more likely that this phase corresponds to primary yellow or brownish aragonite (e.g., Zwicker et al. 2015; Marine and Petroleum Geology 66, 616-630).

**Authors:** That was our first idea, but with high magnification micrographs evidencing the presence of brown filaments in a clear background, we shifted to the filament-based rather than iron-based color. We will reexamine this hypothesis in the line of the references provided and this comment.

**Referee 1:** (12) Lines 554 to 572: This is where authigenic silica formation and silicification at seeps needs to be discussed (see general comment).

**Authors:** OK

**Referee 1:** (13) Chapter 5.3.3.: This chapter does not add much to the manuscript – the discussion is vague to say the least.

**Authors:** OK

**Referee 1:** (14) Line 607: "burrowers feeding on chemosymbiotic microbial communities" – 'Chemosymbiosis' refers to the association of a metazoan host (e.g., bivalve,

tubeworm) with endosymbiotic, chemotrophic bacteria. The term 'chemosynthetic' would work in this instance.

**Authors:** OK, thanks

**Referee 1:** (15) Lines 609 to 610: See also Zwicker et al. (2015, see above) for the role of burrows as part of the shallow plumbing systems in sediments affected by seepage.

**Authors:** OK

**Referee 1:** (16) Line 640: "methane generation zone" – This should be replaced by 'methanic sediments' (i.e., sediment containing methane). Methanogenesis (i.e., methane formation) occurs at greater depth, although minor methanogenesis may also occur at or close to the sulfate-methane transition zone.

**Authors:** OK

**Referee 1:** (17) Fig. 7 (E) and (F): Could this be Beauvoisina carinata (see comment 4)? The shell seems pretty asymmetric, maybe more asymmetric than in B. carinata.

**Authors:** We will check this point

**Referee 1:** Technical corrections and suggestions:

**Authors:** Many thanks to the reviewer for taking the time to spot and report all these mistakes, there is not much to comment or respond to here, we will just correct as requested in the resubmission.

**Referee 1:** (1) Line 26 and throughout: omit blank between numbers and per mil symbol.

**Authors:** OK

**Referee 1:** (2) Line 30: "post-dating the burial" is an ambiguous formulation. It could be misinterpreted in the sense that this phase formed after uplift during telogenesis.

**Authors:** OK

**Referee 1:** (3) Line 31: "late final blocking" – I do not understand. Do you mean that the fluid pathways have been plugged at some point?

**Authors:** What we interpret as fluid pathways (tubular structures) are now (at the time of studying the outcrop) filled with cements, with the dominant final mineral phase in a

burrow being saddle dolomite. Assumig temperature equilibrium between circulating fluids and the host formation, saddle dolomite (the "final blocking phase") would have precipitated when the pseudobioherm was buried to an ambient temperature of 60-80°C at least. The tubes must then have remained open until this temperature was reached by burial, i.e. on the order of magnitude of 1 km (hence the term "late"), which we interpret to largely exceed the initial thickness of the pseudobioherm.

**Referee 1:** (4) Line 45 and 46, and throughout: "H2S and HS" – Why do you use formula instead of words? Before you used the word methane and not its formula. Be consistent. If chemical formula are used, charges (HS-) need to be indicated too.

**Authors:** OK

**Referee 1:** (5) Line 65: improve wording

**Authors:** OK

**Referee 1:** (6) Line 94: 'implies' instead of 'imply'

**Authors:** OK

**Referee 1:** (7) Line 141: "PBH's (pseudobioherms) – You use many abbreviations and acronyms; this does not make reading any easier. What is the benefit of replacing the word 'pseudobioherms' by the abbreviation 'BHPs'? Saving space? Consider to refrain from introducing yet another new abbreviation

**Authors:** we will!.

**Referee 1:** (8) Line 144: Consider to use 'perimeter' instead of 'circle'.

**Authors:** OK

**Referee 1:** (9) Line 171: Same as for per mil. Omit blank between numbers and per cent symbol.

**Authors:** OK

**Referee 1:** (10) Line 246: Add blank before "As".

**Authors:** OK

**Referee 1:** (11) Line 257, and throughout: It is 'gray' in American English.

**Authors:** OK

**Referee 1:** (12) Line 261: add blank after "of"

**Authors:** OK
**Referee 1:** (13) Line 268: "fabric is " not "fabricis"
**Authors:** OK
**Referee 1:** (14) Line 294: "burrows have" not "burrow shave"
**Authors:** OK
**Referee 1:** (15) Line 296: add blank after "Burrow"
**Authors:** OK
**Referee 1:** (16) Line 310: add blank after "10"
**Authors:** OK
**Referee 1:** (17) Line 313: rather "Taken together"
**Authors:** OK
**Referee 1:** (18) Line 383, and throughout: "main silicification surface (MSS)" – the use of such abbreviation impedes comprehensibility.
**Authors:** OK
**Referee 1:** (19) Line 410: "measurements" not "mearurement" when "are" is used.
**Authors:** OK
**Referee 1:** (20) Line 412: blank after per mil symbol.
**Authors:** OK
**Referee 1:** (21) Line 412: "pole" – I do not understand. Do you mean 'pool'? But even than such wording would be less than ideal.
**Authors:** Sorry, that was Frenglish. The term "pole" is commonly used in French for pure phases in a ternary phase diagram. That will be corrected with the help of a native speaker (end-members of a mixing trend?):....
**Referee 1:** (22) Line 414: blank after "and"
**Authors:** OK
**Referee 1:** (23) Line 422: insert blank after "limited".
**Authors:** OK
**Referee 1:** (24) Line 467: "whereas" instead of "where as".
**Authors:** OK

**Referee 1:** (25) Line 467, and throughout: "depleted values" – Colloquial wording. What is it, a value would be depleted in? A values is a number; in this case 'low values' or 'negative values' would be appropriate. A mineral phase, for example, can be depleted in one isotope, in this case C-13, but not a value.
**Authors:** OK
**Referee 1:** (26) Line 486: delete "depletion"
**Authors:** OK
**Referee 1:** (27) Line 492: add blank before "of".
**Authors:** OK **Referee 1:** (28) Line 501: insert blank before "signature" .
**Authors:** OK
**Referee 1:** (29) Line 531: What is "biodeformation"? Is this a good term?
**Authors:** Actually, there are lots of references for "biodeformational structures", defined by Wetzel (1991) as follows: "In general, two types of bioturbation structures were distinguished as suggested by Schäfer (1956); trace fossils ("distinct burrows" sensu Frey and Wheatcroft, 1989), which show a defined shape and have sharp and distinct outlines allowing classification in terms of paleontological nomenclature (Häntzschel, 1975), and biodeformational structures ("indistinct burrows" or "burrow mottles" sensu Frey and Wheatcroft, 1989), which have indistinct outlines and features which destroy pre-existing structures". Biodeformation was meant to be the process responsible for biodeformational structures. There are also references to biodeformation, e.g. Virtasolo et al. Sedimentology 2011. Basically, biodeformation refers to reworking by organisms of sediment not consolidated enough for the burrows to have a lear expression, i.e. the shallowest tier(s) of bioturbation. We will either replace the term by a periphrase (?) (Linguee suggests "paraphrase",;but this is not what I mean; anyway…) or provide appropriate references, we'll make sure anyway that the message is made clear.
**Referee 1:** (30) Line 599 to 600: "… starting from the top shallow within the seafloor" – improve wording.
**Authors:** OK

**Referee 1:** (31) Fig. 10 (B) seems out of focus.
**Authors:** This is UV fluorescence, and diffusion of light emitted within the sample likely blurs the image irrespective of focusing issues.
**Referee 1:** (32) Line 1041: Add full stop after 'side'. (33) Fig. 12 (D) seems out of focus.
**Authors:** this is very high magnification, and it was not possible to have a perfect focus over the whole 30-$\mu$m thickness of the thin section.
**Referee 1:** (34) Line 1093: 'gray' in American English.
**Authors:** OK
**Referee 1:** (35) Line 1108: add blank between numbers and units.
**Authors:** OK
**Referee 1:** (36) Line 1111: '13' in superscript.
**Authors:** OK
**Referee 1:** comments are also provided in a pdf document I will upload Please also note the supplement to this comment: https://se.copernicus.org/preprints/se-2020-187/se-2020-187-RC1-supplement.pdf
**Authors:** OK

---

## Author Comment (AC2) · 13 Apr 2021

**Answers to comments of Referee 2**

**Referee 2:** General comments: The manuscript describes a Jurassic seep carbonate body cropping out in the Aurel area (SE France basin) and focuses particularly on the control exerted by the bioturbation on the vertical growth of the carbonate body. Authors interpret that intense burrowing by callianassid-type shrimps in the central part of the seep enhanced vertical permeability during a long time, which avoided the

self-sealing process in the seep deposits and allowed the vertical aggradation of the carbonate body. This work contributes to a better understand on the sedimentation in seep environments and particularly on the formation of high aggrading carbonate bodies. I find this work interesting and it adds to the knowledge about seep-related processes and products. Therefore, I recommend its publication after moderate to major revision. In the manuscript, I find particularly well described, interpreted and discussed the sedimentary facies architecture and C isotopes. Nevertheless, I have two major general comments about the origin of the tubular structures and burrowing.

**Authors:** Thanks for this precise and detailed review. The most significant change will probably be the transfer of the section that concerns bioturbation from results (where we thought it was well-established enough to belong) to discussion, i.e. point 1 below. Other points are mostly matters of clarification (figures/wording/reasoning/etc.).

**Referee 2:** 1) Origin of tubular structures: It is presented the tubular structures within the seep carbonates as biogenic, e.g. trace fossils, but a discussion about other possible origins (abiogenic gas conduits) is missing. In this regard, abiogenic conduits have been well documented in the literature, both in present-day and ancient seeps, and some of their complex networks resemble that of the Aurel pseudobioherm. In addition, I find the interpretation as burrows should be supported on more data or evidences (the only macroscopic sample presented correspond to a single 11 cm-long rock fragment) and then a discussion on the origin would be relevant.

**Authors:** There are two comments here: one about the fact that the macroscopic interpretation is based on a single sample (actually, two of them are figured, respectively in Fig. 8 and Fig. 9+10). We will provide more, either as an extra figure, or possibly as supplementary material to avoid overloading the manuscript. The second comment is about the discussion of bioturbation vs. mechanical (fluid expulsion) cause for the tubular structures. We will develop and discuss this point.

**Referee 2:** 2) Burrowing: Burrows are classified in three size categories (large, medium and small) each of which is interpreted (based on crosscutting relationships) as formed in three consecutive phases at progressively deeper tiers. However,

manuscript's Figure 8 shows that large burrow (Ba) contains centered medium burrows (B1-B3) and they present a parallel and no cross-cutting relationship. It seems, at least from that figure, that medium burrows are actually cement-filled holes of the large burrow and not different burrows.

**Authors:** This is why we included the sample shown in Figs. 9-10. The sample in Fig. 8 is the one we found clearest in showing large burrows filled with peloids (later interpreted as pellets), clearest in our interpretation because it underwent a single population of large burrows. Other samples (e.g. Figs 9-10) have a much more complex distribution of micrite and grainstone patches, which we interpret to an increased degree of burrowing with many intersecting burrows all filled with similar peloids, and likely with micrite intraclasts. The latter set would represent remnants of the initial sediment homogenized by bioturbation at a soupy, unconsolidated stage.

**Referee 2:** Nor do the figures show a clear crosscutting relationship between small and larger burrows.

**Authors:** Unfortunately, none of the sample photos/photomicrographs provide clear cross-cutting evidence. Our interpretation is deroved in good part from the similarity of the observed texture with that of a seep carbonate sample figured in Wetzel, A. (2013). Formation of methane-related authigenic carbonates within the bioturbated zone—An example from the upwelling area off Vietnam. Palaeogeography, palaeoclimatology, palaeoecology, 386, 23-33. We will revisit in detail all available material, illustrate the best way we can what is firmly established, and state explicitly what analogies support more tentative interpretations.

**Referee 2:** I think that this is a key point to interpret the temporal and spatial (depth) distribution of the burrows and, therefore, cross-cutting relationships among burrows should be better illustrated or with more figures (they could be in the "supplementary material").

**Authors:** Fair enough, we will indeed consider the "supplementary material" option since there are already 17 figures.

**Referee 2:** Specific comments: Lines 201: Show the three units in Figure 3.

**Authors:** OK

**Referee 2:** Line 280. It is not clear in Fig. 8 that medium burrows cut through large burrows.

**Authors:** OK, cf. response above

**Referee 2:** Line 289. Indicate figure (Fig. 8?). t3 in Figure 8 is too small to observe concentric bioclast orientation.

**Authors:** Sorry, T3 is actually a typo for B3. The figure is probably too small for the bioclasts to be clearly visible, we have a good photomicrograph showing this point, which we will include (as supplementary material if needed).

**Referee 2:** Lines 313-315: Add reference.

**Authors:** OK

**Referee 2:** Line 324. Smooth wall character does not indicate that it be Trypanites but other criteria as cut bioclasts, etc.

**Authors:** Indeed, but difficult to image in the very fine-grained, bioclast-poor lithology. We actually draw an analogy to the example shown by Wetzel, A. (2013). Formation of methane-related authigenic carbonates within the bioturbated zone—An example from the upwelling area off Vietnam. Palaeogeography, palaeoclimatology, palaeoe- cology, 386, 23-33. The adverb "likely" acknowledges the absence of firm evidence.

**Referee 2:** Line 335-336. Why does homogeneous micrite-rich fabric reflects high bioturbation if there is no evidence of burrowing? Why sediment homogenization or mixing could not be due to other process, as for example gas bubble ascending?

**Authors:** We will discuss this point, going back to bibliography as needed.

**Referee 2:** Line 354: Description of microfacies and diagenesis (section 4.5) is organized in tiers 1, 2, and 3, but these tiers are interpretative, and interpretations should be located in the discussion. Therefore, I recommend to delete them from that section.

**Authors:** That is a significant change in paper architecture, we ended the manuscript being confident that sediment texture was convincing enough for their interpretation as burrows to be considered a result; this is clearly not the case, this point will be moved

to discussion and the "tier" terminology be replaced by non-interpretive terms.

**Referee 2:** Moreover, description of carbonate phases would be more understandable if they will be presented following cement stratigraphy.

**Authors:** We may add a bar diagram of the paragenetic sequence, this should help.

**Referee 2:** In general, I miss comparison with and references to other papers on seep carbonates and particularly on paragenetic sequences.

**Authors:** OK

**Referee 2:** Line 356: What was bioturbated, the original marl or the later micrite carbonate?

**Authors:** Sorry, the wording was not clear. Micrite is used in a too loose manner for both pure lime micrite (deposited as limestone beds) and for cemented marls (shaly limestone, which does not clarify the matter.

**Referee 2:** Line 397. It is used in this line and through the text "synsedimentary cements" (also "sedimentary cements") to indicate early diagenetic cements. I would be better to use always "early diagenetic cements" in contrast to "late diagenetic cements".

**Authors:** OK

**Referee 2:** Line 495: Most D18O values (table in Appendix I and Fig. 13A) corresponding to saddle dolomites are about -1 to -2 per mil. These values are very strange, are higher than reference sediments and early diagenetic cements, and they are not compatible with hot fluids (>60-80 °C) from which saddle dolomites precipitate. Common D18O values for saddle dolomites documented in the literature are around -6 per mil or lower. It needs some discussion.

**Authors:** OK

**Referee 2:** Line 575: "sediment-cement alternations" change by "sediment-cement sequences". Always use the same terminology for the same things.

**Authors:** OK

**Referee 2:** Line 593: Add reference to Fig.3 (". . .A and B; Fig.3").

**Authors:** OK

**Referee 2:** Line 597: It is mentioned the downward growth of concretionary crusts. However, it is not clear whether this interpretation corresponds only to layers A and B or to the entire pseudobioherm. It should be state more explicitly.

**Authors:** To A and B only, this will be written explicitly.

**Referee 2:** Line 605: It is mentioned that the axis of vertically stacked carbonates shows two lateral shifts coinciding with marker beds A and B. Then, these shifts are interpreted that hydrocarbon-charged fluids migrated upslope. However, Fig. 3B suggests that the axis of the PBH migrated in opposite directions, first westward and then eastward. How can this apparent contradiction be explained?

**Authors:** Fig 3D indicates a ca. 5 times higher northward than westward shift across marker bed B. What is shown in 3B is thus an apparent shift. Moreover, the axis is drawn as a best guess to illustrate a visual perception and cannot be defined from the outcrop with the precision suggested by the thin dash-dot line. The text will be revised accordingly.

**Referee 2:** What was the regional and/or local paleoslope orientation at Middle Callovian times?

**Authors:** Actually, there is no clue in this area where outcrop continuity is limited, and that has been subjected to several orogenic phases so that regional geology cannot help as regards the local setting, the one that influences local bubble migration. The interpretation we propose is the simplest we can think of, based on Casenave et al., 2017, which observes this type of upslope shift on a present-day slope offshore W Africa. We will refer to this paper.

**Referee 2:** Line 608: Change "chemosymbiotic microbial communities" by "chemosyn-thetic microbial communities".

**Authors:** OK

**Referee 2:** Figure 8: The legend of this figure is fragmentary and very complex with a lot of symbols, colors (not easily identifiable), etc. It should be a single and simpler legend.

**Authors:** OK

**Referee 2:** Technical corrections: - Omit blank spaces front and back "/" and "-"symbols. Revise throughout the document. - Omit blank spaces between number and °C symbol. Revise throughout the document. - Omit blank space between number and per mil and per cent symbols. Revise throughout the document. - Insert blank space between two words. In many places of the text, blank spaces between words are missing. Revise throughout the document.

**Authors:** Thanks for these, the manuscript will be corrected accordingly.

---

## Author Response (AR1)

General comments I have carefully read this manuscript and found it to be of interest. The manuscript focuses on the role of bioturbation in creating fluid pathways at methane seeps. Burrows, in particular those of decapod crustaceans, are suggested to favor the vertical aggregation of seep deposits despite of the self-sealing effect of carbonate crust formation. By providing a detailed description of the burrow network of a Jurassic seep deposit, this works adds to the literature on seep environments. Two of my comments are of a more general nature.

*Our responses are highlighted in green after each specific point*

(1) It is a missed chance that silicification, formation of chalcedony, and precipitation of euhedral quartz crystals are not put into perspective with the same phenomena [at] other seeps described in the literature. Like for the studied Aurel seep deposit, silicification and silica authigenesis have been observed to postdate the precipitation of methane-derived carbonate, but to predate later diagenetic phases lacking C-13 depletion. This relationship has now been described for many seep deposits and a hypothesis to explain the observed paragenetic sequence has been developed (Kuechler et al., 2012, Lethaia 45, 259-273; Smrzka et al. 2015, Palaeogeography Palaeoclimatology Palaeoecology 420, 13-26). Discussing the context of silicification and silica authigenesis will help to elaborate the postulated timeline of events.

*We have now discussed the issue in more detail (end of section 5.4.2, l. 580-600), but the association of the main phase of silicification with high-temperature saddle dolomite suggests that the main silicification episode post-dates AOM demise. On the other hand, the hypothesis developed by Kuechler and Smrzka could well account for the less pervasive early silicification episode.*

(2) It is mentioned that carbonate crusts may grow downward at seeps (Bayon et al. 2009). Yet possible implications of downward aggregation are not discussed. If, indeed, seep deposits will preferentially grow downward into the sedimentary column, the impact of bioturbation on maintaining fluid flow on longer time scales will be more limited than suggested in this manuscript. Based on the study of mesofabrics of seep limestones, it had been suggested that the aggregation of methane-derived carbonate may proceed downward (Greinert et al. 2002, Int. J. Earth Sci. 91, 698-711; Peckmann et al. 2002, Sedimentology 49, 855-873). While downward growth may indeed occur, the work of Liebetrau et al. (2014, Int. J. Earth Sci. 103, 1845-1872) suggested that upward growth is typically more pronounced. These findings, particularly the work of Liebetrau and co-workers, should be discussed and their relevance should be put into perspective to the inferred role of bioturbation in the formation of the Aurel seep deposit. The authors seem to suggest preferential upward aggregation in case of the Aurel deposit, but this needs to be clarified and should be discussed in more depth.

*We have rewritten the discussion in section 5.5 and tried to make it clearer than in the previous version of the manuscript..*

I found it difficult to follow the captions of the figures with photomicrographs. Figure captions should be self-explanatory on the one end – these are not – and should be succinct on the other end – which they are not either. Consider to focus on what is really needed for the description of the micrographs and what can be moved to the main text. The paragenetic sequence should be apparent from the caption itself.
*We have done our best to simplify and focus.*

The authors manage to get the message across, but the standard of the English is less than ideal. The manuscript would benefit from linguistic editing of a native speaker.

**Authors' answers to referees**

In conclusion, I recommend publication of this interesting work after moderate to major revision.

A brand new publication that should be considered during revision: Gay et al. (2020) Poly-phased fluid flow in the giant fossil pockmark of Beauvoisin, SE basin of France. BSGF-Earth Sciences Bulletin 2020, 191, 35. *Read and cited where appropriate*

Specific comments on the manuscript Note: I do not use special characters in this web-based review

(#1) Line 45: I do not want to be nit-picking, but the precipitation of dolomite requires magnesium ions in addition to calcium. *Fixed*

(#2) Line 141: Chemosymbiosis can only be assumed in case of ancient taxa *Fixed*.

(#3) Lines 239 to 240: What would be an "altered peloid" – please specify. *Fixed*

(#4) Line 264: The work of Rolin et al. (1990) is not the latest publication on the Beauvoisin lucinids. A new species has been formally described by Kiel et al. (2010; Zootaxa 2390, 26-48). *Thank you, Fixed*

(#5) Result chapter, petrography (e.g. page 10): The circumstance that the mineral phases of the paragenetic sequence are not described in chronological sequence impedes comprehensibility.
*We have now replaced the "main silicification surface (MSS) / below MSS / above MSS order, which was admittedly an unnecessary complication by a straight succession: early (pre-silicification) / silicification / late (post-silicification).*

(#6) Chapter 5.2: The reasoning about carbon stable isotopes is mostly okay. Yet, based on the carbon stable isotope data alone, a relation to methane seepage cannot be proven. The described limestone deposit should be compared with the nearby Beauvoisin seep deposits, for which the involvement of anaerobic oxidation of methane in carbonate formation has been proven with lipid biomarkers. *Fixed*

(#7) Lines 509 to 510: "limestone column" – The sedimentary strata do not consist of limestone only. *Fixed with limestone-dominated*

(#8) Line 533, and throughout: "MDAC" – This abbreviation has not been introduced. But why would you like to use it anyway? 'Seep carbonates' are one type of 'methane-derived authigenic carbonates'. Carbon-13 depleted phases of septarian concretions are another example. The designation 'seep carbonate' is consequently more specific than the acronym 'MDAC'. *MDAC replaced by "seep carbonate" throughout the manuscript*

(#9) Line 535: Silicification predates the formation of burial cement. I would not call such silicification 'late', although it is admittedly later than the formation of methane-derived cement. *"late" removed*

(#10) Line 537: "calcite precipitation" – You cannot exclude that much of the calcium carbonate precipitated as aragonite cement like at most modern and Phanerozoic seeps. *.."calcite" replaced with "calcium carbonate"*

(#11) Line 543: "brown color of BM spar – Based on its position in the paragenetic sequence, I consider it more likely that this phase corresponds to primary yellow or brownish aragonite (e.g., Zwicker et al. 2015; Marine and Petroleum Geology 66, 616-630). *That was our first idea, based on previous study of seep carbonate bodies less affected by diagenesis. However, we find it difficult to reconcile the "aragonite" hypothesis with the fact that the brown character is strictly related with filament bushes whose morphology does not appear to match that of published aragonite botryoids.*

(#12) Lines 554 to 572: This is where authigenic silica formation and silicification at seeps needs to be discussed (see general comment) *Done*.

(#13) Chapter 5.3.3.: This chapter does not add much to the manuscript – the discussion is vague to say the least. *Section removed*

(#14) Line 607: "burrowers feeding on chemosymbiotic microbial communities" – 'Chemosymbiosis' refers to the association of a metazoan host (e.g., bivalve, tubeworm) with endosymbiotic, chemotrophic bacteria. The term 'chemosynthetic' would work in this instance. *Fixed*

**Authors' answers to referees**

(#15) Lines 609 to 610: See also Zwicker et al. (2015, see above) for the role of burrows as part of the shallow plumbing systems in sediments affected by seepage. *Zwicker included for this point and others, as appropriate*

(#16) Line 640: "methane generation zone" – This should be replaced by 'methanic sediments' (i.e., sediment containing methane). Methanogenesis (i.e., methane formation) occurs at greater depth, although minor methanogenesis may also occur at or close to the sulfate-methane transition zone. *Fixed*

(#17) Fig. 7 (E) and (F): Could this be Beauvoisina carinata (see comment 4)? The shell seems pretty asymmetric, maybe more asymmetric than in B. carinata. *We finally keep the question mark, the carinate character cannot be observed on the internal mold available, and the shell is definitely more asymmetric than the type samples*

Technical corrections and suggestions
*Many thanks to the reviewer for taking the time to spot and report all these mistakes, there is not much to comment or respond to here, we will just correct as requested in the resubmission.*

(T-1) Line 26 and throughout: omit blank between numbers and per mil symbol. *Fixed*

(T-2) Line 30: "post-dating the burial" is an ambiguous formulation. It could be misinterpreted in the sense that this phase formed after uplift during telogenesis *Fixed*

(T-3) Line 31: "late final blocking" – I do not understand. Do you mean that the fluid pathways have been plugged at some point? *We have changed the wording to insist on the fact that the tubes remained open long after the pseudobioherm was buried rather than on the fact that they were eventually plugged, in a phase largely post-dating all the phenomena discussed in the manuscript.*

(T-4) Line 45 and 46, and throughout: "H2S and HS" – Why do you use formula instead of words? Before you used the word methane and not its formula. Be consistent. If chemical formula are used, charges (HS-) need to be indicated too *Fixed*

(T-5) Line 65: improve wording *Fixed*

(T-6) Line 94: 'implies' instead of 'imply' *Fixed*

(T-7) Line 141: "PBH's (pseudobioherms) – You use many abbreviations and acronyms; this does not make reading any easier. What is the benefit of replacing the word 'pseudobioherms' by the abbreviation 'BHPs'? Saving space? Consider to refrain from introducing yet another new abbreviation *Fixed*

(T-8) Line 144: Consider to use 'perimeter' instead of 'circle'. *Fixed*

(T-9) Line 171: Same as for per mil. Omit blank between numbers and per cent symbol. *Fixed*

(T-10) Line 246: Add blank before "As". *Fixed*

(T-11) Line 257, and throughout: It is 'gray' in American English. *Fixed*

(T-12) Line 261: add blank after "of" *Fixed*

(T-13) Line 268: "fabric is " not "fabricis" *Fixed*

(T-14) Line 294: "burrows have" not "burrow shave" *Fixed*

(T-15) Line 296: add blank after "Burrow" *Fixed*

(T-16) Line 310: add blank after "10" *Fixed*

(T-17) Line 313: rather "Taken together" *Fixed*

(T-18) Line 383, and throughout: "main silicification surface (MSS)" – the use of such abbreviation impedes comprehensibility *Fixed, MSS only appears in figures and is explicated in the caption*

**Authors' answers to referees**

(T-19) Line 410: "measurements" not "mearurement" when "are" is used *Fixed*

(T-20) Line 412: blank after per mil symbol *Fixed*

(T-21) Line 412: "pole" – I do not understand. Do you mean 'pool'? But even than such wording would be less than ideal. *Sorry, that was Frenglish. The term "pole" is commonly used in French for pure phases in a ternary phase diagram. That will be corrected with the help of a native speaker (end-members of a mixing trend?):*...

(T-22) Line 414: blank after "and" *Fixed*

(T-23) Line 422: insert blank after "limited" *Fixed*

(T-24) Line 467: "whereas" instead of "where as" *Fixed*

(T-25) Line 467, and throughout: "depleted values" – Colloquial wording. What is it, a value would be depleted in? A values is a number; in this case 'low values' or 'negative values' would be appropriate. A mineral phase, for example, can be depleted in one isotope, in this case C-13, but not a value. *Fixed*

(T-26) Line 486: delete "depletion" *Fixed*

(T-27) Line 492: add blank before "of" *Fixed*

(T-28) Line 501: insert blank before "signature" *Fixed*

(T-29) Line 531: What is "biodeformation"? Is this a good term? *Wording has been modified and appropriate references added*.

(T-30) Line 599 to 600: "… starting from the top shallow within the seafloor" – improve wording *Fixed*

(T-31) Fig. 10 (B) seems out of focus. *This is UV fluorescence, and diffusion of light emitted within the sample likely blurs the image irrespective of focusing issues*

(T-32) Line 1041: Add full stop after 'side'.

(T-33) Fig. 12 (D) seems out of focus *We have replaced the image by the best quality we could obtain, but there is still a sense of "out of focus". This is probably due at least in part to the high magnification and the fact that it is difficult to have perfect focus on the whole 30-m thickness of the thin section*.

(T-34) Line 1093: 'gray' in American English *Fixed*

(T-35) Line 1108: add blank between numbers and units *Fixed*

(T-36) Line 1111: '13' in superscript *Fixed* comments are also provided in a pdf document I will upload Please also note the supplement to this comment: https://se.copernicus.org/preprints/se-2020-187/se-2020-187-RC1-supplement.pdf

**Authors' answers to referees**

Solid Earth Discuss.,
https://doi.org/10.5194/se-2020-187-RC1, 2020
General comments: The manuscript describes a Jurassic seep carbonate body cropping out in the Aurel area (SE France basin) and focuses particularly on the control exerted by the bioturbation on the vertical growth of the carbonate body. Authors interpret that intense burrowing by callianassid-type shrimps in the central part of the seep enhanced vertical permeability during a long time, which avoided the self-sealing process in the seep deposits and allowed the vertical aggradation of the carbonate body. This work contributes to a better understand on the sedimentation in seep environments and particularly on the formation of high aggrading carbonate bodies. I find this work interesting and it adds to the knowledge about seep-related processes and products. Therefore, I recommend its publication after moderate to major revision. In the manuscript, I find particularly well described, interpreted and discussed the sedimentary facies architecture and C isotopes. Nevertheless, I have two major general comments about the origin of the tubular structures and burrowing.

*Our responses are highlighted in green after each specific point*

1) Origin of tubular structures: It is presented the tubular structures within the seep carbonates as biogenic, e.g. trace fossils, but a discussion about other possible origins (abiogenic gas conduits) is missing. In this regard, abiogenic conduits have been well documented in the literature, both in present-day and ancient seeps, and some of their complex networks resemble that of the Aurel pseudobioherm. In addition, I find the interpretation as burrows should be supported on more data or evidences (the only macroscopic sample presented correspond to a single 11 cm-long rock fragment) and then a discussion on the origin would be relevant.

   *There are two comments here: first about the fact that most of the macroscopic interpretation is based on a single sample. We have now included 3 more samples, and described in more detail the sample of Figs. 8 and 10 (revised numbering), so that the description is now based on 5 macroscopic samples.*

   *The second comment is about the discussion of bioturbation vs. mechanical (fluid expulsion) cause for the tubular structures. This point has now been put as the first subsection of the discussion (section 5.1, alternatives to the bioturbation interpretation).*

2) Burrowing: Burrows are classified in three size categories (large, medium and small) each of which is interpreted (based on crosscutting relationships) as formed in three consecutive phases at progressively deeper tiers. However, manuscript's Figure 8 shows that large burrow (Ba) contains centered medium burrows (B1-B3) and they present a parallel and no cross-cutting relationship. It seems, at least from that figure, that medium burrows are actually cement-filled holes of the large burrow and not different burrows. *This results from our previous choice to show only our "Rosetta sample" that best showed the three sets of burrows.*

**Authors' answers to referees**

*In this sample, a significant subset of burrows appear as observed by the reviewer to occupy a central position in large burrows. Adding other, more typical samples shows that phase-3 burrows are seldom located in the center of peloid limestone patches filling large burrows. Moreover, Thalassiniodes/Spongeliomorpha, well characterized here from the variability of diameter and orientation, never show a central tube in the middle of a 3 times wider actively filled burrow.*

Nor do the figures show a clear crosscutting relationship between small and larger burrows. I think that this is a key point to interpret the temporal and spatial (depth) distribution of the burrows and, therefore, cross-cutting relationships among burrows should be better illustrated or with more figures (they could be in the "supplementary material"). *We have now included additional samples to illustrate the variability of facies F4, with a specific focus on cross-cutting relationships between successive bioturbation phases.*

Specific comments:

**0    Lines 201: Show the three units in Figure 3 *Fixed*.**

**1    Line 280. It is not clear in Fig. 8 that medium burrows cut through large burrows *Fair enough, see response above*.**

**2    Line 289. Indicate figure (Fig. 8?). t3 in Figure 8 is too small to observe concentric bioclast orientation. *Sorry, T3 was actually a typo for B3. We have amended the text so as to correct this point*.**

**3    Lines 313-315: Add reference. *Fixed**

**4    Line 324. Smooth wall character does not indicate that it be Trypanites but other criteria as cut bioclasts, etc. *Here again, the previous description was very confusing in part due to the above-mentioned typo that suggested active fill around interpreted Trypanites. The text has been revised accordingly**

**5    Line 335-336. Why does homogeneous micrite-rich fabric reflects high bioturbation if there is no evidence of burrowing? Why sediment homogenization or mixing could not be due to other process, as for example gas bubble ascending? *This point is discussed in new section 5.1 with appropriate references.**

**6    Line 354: Description of microfacies and diagenesis (section 4.5) is organized in tiers 1, 2, and 3, but these tiers are interpretative, and interpretations should be located in the discussion. Therefore, I recommend to delete them from that section *One of the co-authors having worked extensively on bioturbation in general, and in particular on burrowing in seep carbonates, we are confident about the interpretation of the well-expressed burrows and the succession of events. We have changed the title of the section from "results" to "results and interpretation", and discuss alternative interpretations (in particular mechanical interaction between, ascending fluids and host sediment) in the "discussion" section*. Moreover, description of carbonate phases would be more understandable if they will be presented following cement *Both reviewers made that remark. We have rewritten the corresponding section, which was admittedly confusing in the previous version. We hope the new organization of section 4.4, paragraph "tier 3" will be easier to follow*. In general, I miss comparison with and references to other papers on seep carbonates and particularly on paragenetic sequences *We have added references further to recommendations by both reviewers*.**

**Authors' answers to referees**

**#7** Line 356: What was bioturbated, the original marl or the later micrite carbonate? *Sorry, the wording was not clear. Shallow homogenization by meiofauna affected the original marl and interstratified calcilutite in the first ca. 10 cm below seafloor, before the sediment was cemented by AOM-mediated carbonate precipitation in the porous network, i.e. while the sediment was still soupy. The last phase of bioturbation (Trypanites) predominantly affects micrite pheno-intraclasts, i.e. cemented residual patches of the homogenized sediment.*

**#8** Line 397. It is used in this line and through the text "synsedimentary cements" (also "sedimentary cements") to indicate early diagenetic cements. I would be better to use always "early diagenetic cements" in contrast to "late diagenetic cements" *Fixed.*

**# 9** Line 495: Most D18O values (table in Appendix I and Fig. 13A) corresponding to saddle dolomites are about -1 to -2 per mil. These values are very strange, are higher than reference sediments and early diagenetic cements, and they are not compatible with hot fluids (>60-80 °C) from which saddle dolomites precipitate. Common D18O values for saddle dolomites documented in the literature are around -6 per mil or lower. It needs some discussion *Discussed the best we could in the last paragraph of section 5.3. Actually, we could only acknowledge the difficulty and follow Peckmann et al. (2003) and Zwicker et al. (2015), who state that "Oxygen isotopes of seep carbonates are more challenging to interpret than carbon isotopes, […] mainly due to the ease of oxygen isotope exchange during diagenesis, as the oxygen pool of waters […] involved in late diagenesis is much larger than their carbon pool ». In addition, the focus of this manuscript is more on sedimentation / bioturbation / early diagenesis that on late diagenesis.*

**#10** Line 575: "sediment-cement alternations" change by "sediment-cement sequences". Always use the same terminology for the same things *Fixed.*

**#11** Line 593: Add reference to Fig.3 (". . .A and B; Fig.3") *Fixed.*

**#12** Line 597: It is mentioned the downward growth of concretionary crusts. However, it is not clear whether this interpretation corresponds only to layers A and B or to the entire pseudobioherm. It should be state more explicitly. *The corresponding section has been entirely rewritten.*

**#13** Line 605: It is mentioned that the axis of vertically stacked carbonates shows two lateral shifts coinciding with marker beds A and B. Then, these shifts are interpreted that hydrocarbon-charged fluids migrated upslope. However, Fig. 3B suggests that the axis of the PBH migrated in opposite directions, first westward and then eastward. How can this apparent contradiction be explained? *Fig 3D indicates a ca. 5 times higher northward than westward shift across marker bed B. What is shown in 3B is thus an apparent shift. Moreover, the axis is drawn as a best guess to illustrate a visual perception and cannot be defined from the outcrop with the precision suggested by the thin dash-dot line. The text has been revised accordingly.* What was the regional and/or local paleoslope orientation at Middle Callovian times? *Actually, there is no clue in this area where outcrop continuity is limited, and that has been subjected to several orogenic phases so that regional geology cannot help as regards the local setting, the one that influences local bubble migration. The interpretation we propose is the simplest we can think of, based on Casenave et al., 2017, which observes this type of upslope shift on a present-day slope offshore W Africa. We will refer to this paper.*

**#14** Line 608: Change "chemosymbiotic microbial communities" by "chemosynthetic microbial communities" *Fixed.*

**Authors' answers to referees**

**15 Figure 8: The legend of this figure is fragmentary and very complex with a lot of symbols, colors (not easily identifiable), etc. It should be a single and simpler legend. *(After reorganization, this has become Fig. 9). Panel D (the one with colors) has been simplified, with two unnecessary categories removed, which should help identification. And the legend has been much simplified and grouped in a single box.**

Technical corrections: - Omit blank spaces front and back "/" and "-"symbols *Fixed*. Revise throughout the document. - Omit blank spaces between number and °C symbol. *Fixed* Revise throughout the document. - Omit blank space between number and per mil and per cent symbols. *Fixed* Revise throughout the document. - Insert blank space between two words *Fixed*. In many places of the text, blank spaces between words are missing *Fixed*. Revise throughout the document.

---

## Author Response (AR2)

[revised manuscript text omitted]

Number: 1    Author: stroe    Subject: Sticky Note    Date: 23/06/2021 09:31:49
4. figure doesn't exist, make sure references to figure 7 panels are correct.

Author: Utilisateur  Subject: Note    Date: 29/06/2021 16:01:13
does exist in updated version

Number: 2    Author: stroe    Subject: Highlight    Date: 22/06/2021 22:03:22

**4.2.4.2. Fabric and bioturbation**

The fabric is best characterized by using combined plane and UV light on cut sections. Macroscopically, most samples show a complex mosaic of patches, which under UV light appear to consist of low-fluorescence micrite and calcarenite exhibiting various levels of fluorescence (Fig. 8). At hand lens magnification, the calcarenitic limestone consists of peloid grainstone having fluorescent cement. The proportions of micrite and calcarenite vary, the former ranging from 20–50%, the latter from 80–50%. Near the transition to facies F3, where micrite is enriched (~50%) it forms continuous domains, whereas calcarenitic material constitutes a network of 2–3 cm-wide patches surrounded by a darker, commonly fluorescent halo (Figs. 8C–D, 9A–C). This texture evidences pervasive bioturbation, with two populations of burrows. Calcarenitic domains 2–3 cm in width are referred to as "large burrows" and 0.5–1.5 cm-sized ones as "medium burrows" in the following. In addition, numerous mm-diameter burrows ("small burrows") filled by fluorescent cements are scattered throughout the section. In calcarenite-rich samples, micrite occurs as low/moderate-fluorescence patches, 1 mm to 2 cm in diameter, angular to subrounded, commonly surrounded by a dark halo. The micrite domains are interpreted as remnants of the initial sediment that were not reworked by burrowers. They are described as so-called 'pheno-intraclasts' floating in the dominant peloid grainstone (Fig. 8?–G), in contrast to true intraclasts defined as reworked particles. The overall fabric of calcarenite-rich areas reflects pervasive bioturbation, including but likely not restricted to the large burrows.

**a) Large burrows**

The margins of large burrows, as defined by their dark halo, are rather irregular. The geometrical arrangement of the large burrows suggests that they constitute a complex branched network (see Cunningham et al., 2012). They are filled with sediment, typically peloid grainstone/wackestone identical to that of the background indicative of passive fill. Large burrows have variable intensity of fluorescence, with peloid grainstone being the most fluorescent. At the transition from F3 to F4, granular/fluorescent material becomes progressively enriched. The more granular/fluorescent patches dominantly crosscut less fluorescent ones. (Fig. 8D)

**b) Medium burrows**

The morphology of the medium burrows is best observed by narrow-spaced parallel sections, on opposite sides of a sawcut (Fig. 9A–B). Some burrows show abrupt changes in direction and/or diameter between closely spaced parallel sections (Fig. 9A–B, D). Burrows commonly show a dark halo around the cement fill, which in some cases can be followed into the sediment, defining a subcircular shape (Fig. 9E). Its lower part is filled with sediment and the upper part with cement. Within the burrows, the contact between sediment within burrows and cement above is planar, with an apparent dip of 0°–30° (Fig. 9E), indicating partially passive fill. Some burrows show laminated sediment fill recording multiphase passive infill. Macroscopically, the lumen of most medium burrows shows a concentric bipartite cement fill with a ca. 1 mm-thick tan, translucent, continuous outer rim around the lumen and a white to yellow/gold final axial fill (Fig. 10A–B). Both the dark halo and the outer rim are brightly fluorescent, in contrast to the low-fluorescent final fill (Figs. 9C, 10A–B). Burrow margins commonly show irregularities that might have resulted from burrow excavation, corrosion by fluids circulating in the burrows, in particular when they contain (hydrogen)sulfide, or local collapse of the burrow top. The distance between

Page: 8

Number: 1    Author: stroe    Subject: Sticky Note    Date: 23/06/2021 09:31:53
5. not all the panels are there. check figure and its references to it.

Author: Utilisateur   Subject: Note    Date: 29/06/2021 16:01:38
panels E-G present in updated version

Number: 2    Author: stroe    Subject: Highlight   Date: 22/06/2021 22:20:04

medium burrows ranges from 2–10 cm with an estimated average of 5 cm. Burrows show subvertical, oblique and subhorizontal segments, without preferential orientation. The abrupt changes of orientation and diameter over short distances and the coexistence of vertical, oblique, and horizontal segments are diagnostic of burrows of the *Thalassinoides – Spongeliomorpha* suite (e.g., Cunningham et al., 2012; Knaust, 2017). These are produced by decapod crustaceans that may penetrate as deep as two meters or more below seafloor (e.g., Sarnthein, 1972; Pemberton and Buckley, 1976). *Thalassinoides* corresponds to smooth-walled burrows, being lined with mucus in soft sediment and unlined in sediment stiff enough to remain open and *Spongeliomorpha* to burrows scratched by body appendages indicating firm sediment (Wetzel and Uchman, 1998). Medium burrows occur preferentially in calcarenite and appear to get around pheno-intraclasts (Fig. 8A–B, 2–G). However, in calcarenite patches, they do not follow specific patterns that could suggest spatial relationship between large and medium burrows (Fig. 8).

**c) *Small burrows**

Small, cement-filled burrows are better visible under UV light. Some of these small burrows are directly connected to medium burrows while most appear isolated in vertical section (Fig. 9A, C, F), but out-of-plane contact with medium burrows appears likely. Their geometry, size, and fill match the characteristics of *Trypanites* representing borings in hard substrate. Small burrows may cut across pheno-intraclasts and/or peloid limestone, or locally follow the contact between the two.

**4.3. Relative timing and tiering**

Based on cross-cutting relationships, it is possible to distinguish successive phases of burrowing evidenced by intensely mixed, homogeneous sediment, overprinted by abundant shallow-produced large *Thalassinoides*, in turn cross-cut by deep, but scarcer and smaller *Thalassinoides* or *Spongeliomorpha*; occasional *Trypanites* borings entrenched into the cemented margins of the latter traces represent the last phase of bioturbation. In addition, the non-fluorescent wackestone occurring in patches has a grain content similar to the background sediment of Facies 1 or to the nodular facies F2 and F3. In contrast to the background sediment, it never shows lamination, nor stratification on the scale of limestone-marl alternations of the host sediment. Nonetheless, the processes that produced layering and lamination must have been active at the pseudobioherm site like in the surroundings. The uniform texture of the wackestone, therefore, reflects efficient mixing of the uppermost soupy/soft sediment by near-surface burrowing organisms producing so-called biodeformational structures that overprint pre-existing structures but do not constitute distinct traces (e.g., Schäfer, 1956; Wetzel, 1991). Consequently, it is possible to distinguish four phases of endobenthic activity recorded by

- tier 1 comprising biodeformational structures homogenizing soft sediment into wackestone (Phase 1),
- tier 2 constituted by burrows emplaced in soft to firm sediment mainly by large decapod crustaceans and filled by peloids (Phase 2),
- tier 3 characterized by burrows produced by smaller decapod crustaceans in sediment so stiff that the tunnels remained open (Phase 3), and occasionally,
- tier 4 solely evidenced by *Trypanites* borings originating at the walls of tier-3 burrows (Phase 4).

**4.4. Microfacies and diagenesis**

*Tier 1: Micrite pheno-intraclasts*

| Number: 1 | Author: stroe | Subject: Sticky Note | Date: 23/06/2021 09:32:04 |

6. not all panels are there

| Author: Utilisateur | Subject: Note | Date: 29/06/2021 16:03:06 |

panels E-G present in updated version

[revised manuscript text omitted]

Page: 27

| | | | |
|---|---|---|---|
| Number: 1 | Author: Utilisateur | Subject: Texte surligné | Date: 04/07/2021 18:52:03 |

| | | | |
|---|---|---|---|
| Number: 2 | Author: stroe | Subject: Sticky Note | Date: 04/07/2021 18:40:35 |

7. Beaudrimont reference is missing, check all references again.

Author: Utilisateur   Subject: Note   Date: 12/07/2021 18:34:11
Ok

| | | | |
|---|---|---|---|
| Number: 3 | Author: stroe | Subject: Sticky Note | Date: 04/07/2021 18:51:17 |

8. add comma after "et al.". make sure you have a consistency throughout. Check all occurrences

| | | | |
|---|---|---|---|
| Number: 4 | Author: stroe | Subject: Highlight | Date: 04/07/2021 18:51:28 |

| | | | |
|---|---|---|---|
| Number: 5 | Author: stroe | Subject: Sticky Note | Date: 04/07/2021 18:51:36 |

9. should this be "platform" in both instances?

Author: Utilisateur   Subject: Note   Date: 12/07/2021 18:34:24
Corrected

| | | | |
|---|---|---|---|
| Number: 6 | Author: stroe | Subject: Highlight | Date: 04/07/2021 18:51:31 |

| | | | |
|---|---|---|---|
| Number: 7 | Author: stroe | Subject: Highlight | Date: 04/07/2021 18:51:40 |

| | | | |
|---|---|---|---|
| Number: 8 | Author: stroe | Subject: Sticky Note | Date: 04/07/2021 18:51:24 |

10. Basin

Author: Utilisateur   Subject: Note   Date: 12/07/2021 18:34:39
Corrected

| | | | |
|---|---|---|---|
| Number: 9 | Author: stroe | Subject: Sticky Note | Date: 04/07/2021 18:39:22 |

11. add "et al." Carefully check all your references once more.

Author: Utilisateur   Subject: Note   Date: 12/07/2021 18:35:10
Done

| | | | |
|---|---|---|---|
| Number: 10 | Author: stroe | Subject: Sticky Note | Date: 04/07/2021 18:52:29 |

12. state unit somewhere

Author: Utilisateur   Subject: Note   Date: 12/07/2021 18:35:17
Done

| | | | |
|---|---|---|---|
| Number: 11 | Author: stroe | Subject: Highlight | Date: 04/07/2021 18:39:26 |

| | | | |
|---|---|---|---|
| Number: 12 | Author: stroe | Subject: Sticky Note | Date: 04/07/2021 18:52:57 |

13. this sentence lacks a verb

Author: Utilisateur   Subject: Note   Date: 29/06/2021 16:31:11
done

| | | | |
|---|---|---|---|
| Number: 13 | Author: stroe | Subject: Highlight | Date: 04/07/2021 18:52:34 |

| | | | |
|---|---|---|---|
| Number: 14 | Author: stroe | Subject: Highlight | Date: 04/07/2021 18:53:00 |

| | | | |
|---|---|---|---|
| Number: 15 | Author: stroe | Subject: Sticky Note | Date: 04/07/2021 18:53:57 |

14. refer to the red star

Author: Utilisateur   Subject: Note   Date: 12/07/2021 18:35:27
Done

| | | | |
|---|---|---|---|
| Number: 16 | Author: stroe | Subject: Highlight | Date: 04/07/2021 18:53:52 |

**Figures**

[Figure]

Figure 1: Geological setting of the studied outcrop. A) Cross-section from the margin to the center of the Dauphinois Basin traversing the Aurel borehole (after Wannesson and Bessereau, 1999, modified). B) Facies map of the Dauphinois Basin for the Callovian (Debrand-Passard, 1984); for the basin facies represented by the Terres Noires Formation (middle Bathonian up to lower Oxfordian) in the central part of the basin isopachs are shown (after Beaudrimont and Dubois, 1977). Outcrops of the Terres Noires Formation (after Artru, 1972) and the location of the major pseudobioherms marked by red stars (after Rolin, 1987). C) Lithologic log of the Aurel area (Flandrin, 1974). The Aurel borehole penetrated the carbonate-dominated Dogger and the marly upper Lias. The pseudobioherms in the Aurel area are located in middle Callovian sediments; the pseudobioherms in the Dauphinois Basin occur in upper Bathonian to middle Oxfordian deposits. D) Geological map of the Aurel area (after Flandrin, 1974, modified). Several pseudobioherms are poorly exposed near the Vaunière ruin; the studied pseudobioherm is located near La Touche farm.

[Figure]

**Figure 2: Lithologic log of the middle Callovian deposits in the pseudobioherm-bearing interval, logged about 20 m to the west of the upper part of the pseudobioherm, not affected by faulting; note marker beds A and B, easily recognizable in both host sediment and pseudobioherm.**

[Figure]

**Figure 3: A) and B), Image and interpretation of the southern flank of the Aurel pseudobioherm; dashed lines represent assumed axis of the pseudobioherm. C) and D), Image and interpretation of the eastern flank of the pseudobioherm. LST = limestone.**

[Figure]

Number: 1          Author: stroe          Subject: Sticky Note          Date: 04/07/2021 18:58:13
15. give the inset frame a color that makes it stand out clearer

Author: Utilisateur   Subject: Note          Date: 29/06/2021 16:33:49
I made it white, 1 pt... hopefully OK

**Figure 4: Background facies F1. A) Typical alternation between marl and limestone showing platy bedding. B) Surface**
1025     **of a slab with radiating burrow (*Gyrophyllites multiradiatus*); inset shows the bivalve *Bositra*, the only common fossil in**
**F1. C) Marl with numerous filamentous structures interpreted as disaggregated *Bositra* shells. D) Limestone showing**
**the same microfossil content as the marl but in microsparitic matrix. E) Limestone consisting of alternating packstone**
**and wackestone laminae. F) Marl with numerous recrystallized microfossils in a micritic matrix; circular objects are**
**interpreted as calcispheres or radiolarians; inset showing a tubular structure interpreted as a sponge spicule.**

1030

[Figure]

**Figure 5: Facies F2, nodular marl. A) Outcrop view. B) Individual nodules and nodule clusters. C) and D) Cross-section of a nodule under natural (C) and UV light (D); darker (C) non-fluorescent (D) vertical parting in the middle is suggestive of previous boundary between two, now clustered nodules.**

1035

[Figure]

**Figure 6: Facies F3, nodular limestone. A) and B) Vertical section under natural (A) and UV light (B); arrows point upward. C) Photomicrograph showing contact between a nodule and matrix (marked yellow dashed line). Grain packing in matrix is denser than in nodule, suggesting cementation of the nodule prior to compaction.**

1040

[Figure]

**Figure 7: Facies F4, massive limestone, and associated fauna. A) Weathered surface showing three bivalves shown in**
**(B) to (D) in detail, apparently belonging to different species. Bivalve with geopetal filling. E) and F) Two valves of**
**the only lucinid found (*Beauvoisina carinata*, Kiel et al., 2010). G) Section of a small gastropod. ) Bedding plane view**
**of *Thalassinoides*; smaller tape measure graduations in mm.**

1045

Page: 33

Number: 1          Author: stroe          Subject: Sticky Note          Date: 04/07/2021 19:01:09
16. remove?

   Author: Utilisateur   Subject: Note          Date: 04/07/2021 19:01:05
   I would keep the mention, it highights the variety of fauna in Facies 4, in contrast with the ocurrence of a single bivalve species in
   other facies (Bositra sp.)

Number: 2          Author: stroe          Subject: Sticky Note          Date: 04/07/2021 19:01:51
17. Not shown. remove.

   Author: Utilisateur   Subject: Note          Date: 04/07/2021 19:01:44
   shown in revised version

Number: 3          Author: stroe          Subject: Highlight   Date: 04/07/2021 19:01:47

   Author: Utilisateur   Subject: Note          Date: 29/06/2021 16:37:05
   shown in updated version

Number: 4          Author: stroe          Subject: Highlight   Date: 04/07/2021 19:01:19

[Figure]

1050

Figure 8: Fabric of intensely bioturbated sediment. A) Natural light and B) UV fluorescence light, Facies F4. White arrows in B) indicate micrite pheno-intraclasts. Note the variability of pheno-intraclasts in roundness and size. C) Natural light and D) UV fluorescence light, transition from F3 to F4. In both samples, medium burrows (MB) appear independent from variations in fabric around them; black rectangles drawn for thin section selection. E) Natural light,

1055    F) UV fluorescence light and G) Line drawing, facies F4, unoriented vertical section. White arrows in G) show points where burrow margin tangentially passes pheno-intraclasts.

Page: 34

Number: 1    Author: stroe    Subject: Highlight    Date: 04/07/2021 19:10:25

Number: 2    Author: stroe    Subject: Sticky Note    Date: 04/07/2021 19:10:20
18. Combine "(" with "panel B)"

Author: Utilisateur    Subject: Note    Date: 12/07/2021 18:36:05
the caption belongs to the NEW fig 8, not to this sample. Fig. 8 has been updated.

Number: 3    Author: stroe    Subject: Sticky Note    Date: 04/07/2021 19:10:33
19. this is presumably the caption of figure 9?

Author: Utilisateur    Subject: Note    Date: 04/07/2021 19:11:16
absolutely! Sorry about the mishap, there was a mistake between two sets of figures

Number: 4    Author: stroe    Subject: Highlight    Date: 04/07/2021 19:12:14

Number: 5    Author: stroe    Subject: Sticky Note    Date: 04/07/2021 19:12:20
20. Figure 9 doesn't have panels C-G??

Author: Utilisateur    Subject: Note    Date: 29/06/2021 16:43:36
in the normal version yes

Number: 6    Author: stroe    Subject: Sticky Note    Date: 04/07/2021 19:11:58
21. Panels F and G are not shown. Arrows in panel E needs mentioning.

Author: Utilisateur    Subject: Note    Date: 29/06/2021 16:43:52
ok in updated file

Number: 7    Author: stroe    Subject: Highlight    Date: 04/07/2021 19:12:06

[Figure]

**Figure 9: Burrow architecture in F4. A) and B), Opposing sides of a sawcut, ca. 3 mm apart; Note 3 types of burrows: Large burrows (one marked as Ba) surrounded by white dashed line; medium burrows, (partly) filled with white-yellowish cement (labelled B1–B6); small burrows (t1–t3) encircled in red. C) UV-light view of the section shown in (A). Yellow lines follow main dark halos, as marked in (A). D) Line drawing of (C) showing with marked burrows and halos; yellow dotted line marking outlines of B1–B6 on opposite section B), highlighting variations in size/orientation across the 3-mm gap. E) Close-up of B6 marked in (A) showing burrow surrounded by dark halo (bold arrows) and half-filled with geopetal sediment; thin white arrows show halos in the geopetal filling.
[Figure]
 Cross-cutting relationships between burrows (UV fluorescence). White circles point at small burrows connected to a medium burrow. White rectangle highlights longitudinal cross-sections of small burrows. Black rectangles drawn for thin section selection.**

1060

1065
* * *
Page: 35

Number: 1          Author: stroe          Subject: Sticky Note          Date: 04/07/2021 19:59:13

22. this is presumably the caption for figure 8?

   Author: Utilisateur  Subject: Note          Date: 29/06/2021 16:44:46
   the captions were written for an updated file that was not taken into account

Number: 2          Author: stroe          Subject: Highlight   Date: 04/07/2021 19:59:06

Number: 3          Author: stroe          Subject: Sticky Note          Date: 04/07/2021 19:59:02

23. there is no panel F that I could detect

   Author: Utilisateur  Subject: Note          Date: 29/06/2021 16:45:00
   there is in the updated file

[Figure]

 **Figure 10: see caption next page**

Page: 36

Number: 1    Author: stroe    Subject: Sticky Note    Date: 04/07/2021 19:59:53
24. not mentioned in the text

Number: 2    Author: stroe    Subject: Highlight   Date: 04/07/2021 19:59:49

**Figure 10: A) and B) Close-up of [3]gs. 8 A, B[1] showing the details of burrow infill. Orange arrows in (A) [2]ark dark halo surrounding the burrow on the left side and expanding on the right side into peloid grainstone[4]PG);[1] yellow arrows mark thin, continuous, milky white cement rim following the dark halo on the left side of the image but the sediment-cement contact on the right side. B) UV fluorescence view of the same section; the blurry appearance results from diffusion of light emitted from fluorescent parts into non-fluorescent ones[5]ht = micrite pheno-intraclasts, EC = early cements, LC = late cements; [6]hite rectangle refers to detail shown in C.[1] Pheno-intraclasts, like nodules in F2 and F3 with low level of fluorescence, in contrast to brightly fluorescent peloid grainstone (less fluorescent spots therein are individual peloids). The brightly fluorescent rim coincides with the cement ring limited on either side by an outer dark rim and an inner white rim. C) Microphotograph of the same sample; dark halo [7]range arrows) corresponds to superposition of brown and clear microsparite layers; brown microspar[8]MSpar) in direct contact with burrow wall. The white rim (yellow arrows) consists of beige chalcedony (for details, see Fig. 11). White arrows point to shell fragments, PW = peloid wackestone, Dol 2 = late saddle dolomite. D) and E) Photomicrographs of peloid grainstone in F4. D) Internal texture of the peloids in plane polarized light. E) as (D) in UV epifluorescent [9]ght.**

1075

1080

1085
* * *
**Page: 37**

| Number: 1 | Author: stroe | Subject: Sticky Note | Date: 04/07/2021 20:00:01 |
|---|---|---|---|

25. depending on the numbering of figures 8 and 9

Author: Utilisateur Subject: Note Date: 29/06/2021 16:49:37
correct on new figures

| Number: 2 | Author: stroe | Subject: Sticky Note | Date: 04/07/2021 20:00:36 |
|---|---|---|---|

26. not shown in panel A

Author: Utilisateur Subject: Note Date: 12/07/2021 18:36:30
modified

| Number: 3 | Author: stroe | Subject: Highlight | Date: 04/07/2021 20:00:05 |
|---|---|---|---|

| Number: 4 | Author: stroe | Subject: Highlight | Date: 04/07/2021 20:00:33 |
|---|---|---|---|

| Number: 5 | Author: stroe | Subject: Sticky Note | Date: 04/07/2021 20:00:51 |
|---|---|---|---|

27. this is true for panels A and B

| Number: 6 | Author: stroe | Subject: Highlight | Date: 04/07/2021 20:00:55 |
|---|---|---|---|

Author: -S- Subject: Sticky Note Date: 12/07/2021 18:37:34
Done

Author: -S- Subject: Sticky Note Date: 12/07/2021 18:37:47

| Number: 7 | Author: stroe | Subject: Sticky Note | Date: 04/07/2021 17:50:51 |
|---|---|---|---|

28. not shown in the figure

Author: Utilisateur Subject: Note Date: 29/06/2021 16:56:05
modified

| Number: 8 | Author: stroe | Subject: Highlight | Date: 04/07/2021 17:50:54 |
|---|---|---|---|

| Number: 9 | Author: stroe | Subject: Inserted Text | Date: 04/07/2021 17:47:10 |
|---|---|---|---|

ce

Author: Utilisateur Subject: Note Date: 29/06/2021 16:56:44
done

[Figure]

Figure 11: Cements within Phase-3 burrows and host sediment. A) and B) General view of burrow margin in plane light (A) and cross-polarized light (B). Note continuous rim of beige chalcedony (Chal-2) associated with saddle dolomite (Dol-1). C) Detail of relationship between Chal-2 and carbonate cements in cross-polarized light; hite rectangle. Chal-2 cutting across a single sparite crystal. D) Two stacked sediment-cement sequences in plane polarized light showing alignment of euhedral quartz Q roughly l to burrow margin wall, as well as radiaxial gray calcite crystals (RAx) locally capping d-BMSpar-CMSpar succession; blue zigzag line marks upper limit of euhedral quartz. E) and F), main phases of silicification in plane light (E) and cross-polarized light (F); Chal-1 = botryoidal chalcedony precipitated freely at the tube wall, Q = euhedral quartz also overgrown on a free surface, Chal-2 and the associated saddle dolomite Dol-1 replaced both pre-existing carbonates and Chal-1. Note the extinction pattern of Chal-2 characterizing "flamboyant chalcedony". The blue zigzag line marks the main silicification surface. For details see text.

Page: 38

Number: 1    Author: stroe    Subject: Sticky Note    Date: 04/07/2021 16:41:56
29. can't find

Author: Utilisateur  Subject: Note    Date: 29/06/2021 16:57:45
visible on new version

Number: 2    Author: stroe    Subject: Highlight   Date: 04/07/2021 16:41:52

Number: 3    Author: stroe    Subject: Sticky Note    Date: 04/07/2021 16:42:59
30. explain

Author: Utilisateur  Subject: Note    Date: 04/07/2021 16:42:52
caption of D rewritten in full

Number: 4    Author: stroe    Subject: Highlight   Date: 04/07/2021 16:43:07

Number: 5    Author: stroe    Subject: Sticky Note    Date: 04/07/2021 17:02:51
31. Make sure all the abbreviations are explained when they first occur in panels A-F.

[Figure]

**Figure 12: Microphotographs showing details of sediment-cement sequences. A) Repeated sequence of sediment (sed), brown microsparite (BMSpar), and clear microsparite (CMSpar), locally overprinted by chalcedony (Chal-2). B) Detail of BMSpar-CMSpar contact marked by black dashed line; CMSpar exhibiting filamentous "bushes". C) as (B) in cross-polarized light, showing individual microspar crystal, straddling across BMSpar/CMSpar boundaries (marked by black arrows); white rectangle marks area shown in detail in (D). D) close-up of filamentous bush; note elongate brown inclusions within clear crystal. E) and F) BMSpar/CMSpar contact in plane light (E) and UV epifluorescent light (F); fluorescent inclusions near the margin of BMSpar following oblique contact between BMSpar and CMSpar into the thin section. G) and H) Typical CMSpar in plane light (G) and UV epifluorescent light (H), fluorescence originating from both fluid inclusions (bright white) and microspar (dull yellow).**

Page: 39

Number: 1    Author: stroe    Subject: Sticky Note    Date: 04/07/2021 16:40:23
32. this is not a proper sentence

Author: Utilisateur   Subject: Note    Date: 04/07/2021 15:59:46
corrected

Number: 2    Author: stroe    Subject: Sticky Note    Date: 04/07/2021 16:40:07
33. not that contact: contact with Sed: rephrase

Author: Utilisateur   Subject: Note    Date: 04/07/2021 15:42:34
quite right! rephrased

Number: 3    Author: stroe    Subject: Highlight   Date: 04/07/2021 15:42:35

Number: 4    Author: stroe    Subject: Highlight   Date: 04/07/2021 15:59:31

Number: 5    Author: stroe    Subject: Sticky Note    Date: 04/07/2021 16:40:16
34. this is part of (B) not (C).

Author: Utilisateur   Subject: Note    Date: 04/07/2021 15:57:19
right, modified

Number: 6    Author: stroe    Subject: Sticky Note    Date: 04/07/2021 16:40:30
35. mention the white rectangle!

Author: Utilisateur   Subject: Note    Date: 04/07/2021 15:57:37
done

Number: 7    Author: stroe    Subject: Highlight   Date: 04/07/2021 15:57:39

Number: 8    Author: stroe    Subject: Highlight   Date: 04/07/2021 15:57:22

Number: 9    Author: stroe    Subject: Inserted Text    Date: 23/06/2021 09:00:39
ce

Number: 10    Author: stroe    Subject: Inserted Text    Date: 04/07/2021 20:06:20
ce

1105

1110

[Figure]

1115 **Figure 13: Stable C and O isotopes and microfacies. A) Sedimentary and diagenetic carbonate phases encountered in the Aurel pseudobioherm in δ$^{13}$C-δ$^{18}$O plot; position of samples of facies F1 to F3 is shown in Fig. 14.**  **abric and microfacies of F4 recording successive phases of bioturbation. C) Schematic geometric relationships between sediment and various cement generations**  **within facies F4; black boxes mark**
1120 **minimum size of samples drilled for isotope analysis, some likely mixing different phases like peloid grainstone and BMSpar/CMSpar.** **T = grainstone, MSS = main silicification surface,** details, see text.

1125

40

Page: 40

Number: 1    Author: stroe    Subject: Highlight    Date: 04/07/2021 16:01:09

Number: 2    Author: stroe    Subject: Sticky Note    Date: 04/07/2021 20:07:02
36. Nodules are circles, not ellipses: adjust.

Author: Utilisateur    Subject: Note    Date: 04/07/2021 16:01:01
done

Number: 3    Author: stroe    Subject: Sticky Note    Date: 04/07/2021 20:06:59
37. put white labels behind -2, -4, -10 and -12.

Author: Utilisateur    Subject: Note    Date: 04/07/2021 16:08:16
done for all labels (both axes)

Number: 4    Author: stroe    Subject: Highlight    Date: 04/07/2021 16:08:24

Number: 5    Author: stroe    Subject: Sticky Note    Date: 04/07/2021 16:09:54
38. Not shown in figure

Author: Utilisateur    Subject: Note    Date: 04/07/2021 16:09:51
now shown

Number: 6    Author: stroe    Subject: Cross-Out    Date: 23/06/2021 09:08:49

Number: 7    Author: stroe    Subject: Sticky Note    Date: 04/07/2021 16:21:28
39. mention white box

Number: 8    Author: stroe    Subject: Highlight    Date: 04/07/2021 16:11:46

Number: 9    Author: stroe    Subject: Sticky Note    Date: 04/07/2021 16:13:04
40. not shown in panel C

Author: Utilisateur    Subject: Note    Date: 04/07/2021 16:13:03
removed

Number: 10    Author: stroe    Subject: Cross-Out    Date: 23/06/2021 09:11:47

Author: Utilisateur    Subject: Note    Date: 04/07/2021 20:08:03
removed

Number: 11    Author: stroe    Subject: Sticky Note    Date: 04/07/2021 16:21:11
41. please explain all the acronyms

Author: Utilisateur    Subject: Note    Date: 04/07/2021 16:21:01
all acronyms explained

Number: 12    Author: stroe    Subject: Highlight    Date: 04/07/2021 16:21:04

[Figure]

**Figure 14: Change in depth of the sulfate methane transition zone (SMTZ) and methane-derived authigenic carbonates**
1130 **(brown) from host sediment towards the pseudobioherm; vertical dashed line marking the axis of the pseudobioherm,**
**symbols referring to the isotope data shown in Fig. 13A.  Not to scale.**

Page: 41

Number: 1          Author: stroe          Subject: Highlight   Date: 04/07/2021 16:39:50

Number: 2          Author: stroe          Subject: Sticky Note          Date: 04/07/2021 16:39:42
42. The "shrimp" is not explained in Figure 13A

Author: Utilisateur  Subject: Note          Date: 04/07/2021 16:38:27
"Shrimp" explained, legend expanded to include burrows

Page: 42

Number: 1     Author: stroe     Subject: Inserted Text     Date: 04/07/2021 16:39:21

Author: Utilisateur  Subject: Note     Date: 04/07/2021 16:39:17
space added

[Figure]

1135

Figure 15: Fluid flow in and around the pseudobioherm (PBH). A) Typical architecture of a recent decapod crustacean burrow (*Callianassa truncata,* after Ziebis et al., 1996, modified); gray surfaces indicating hypothetical limits between bioturbation tiers 1 to 3 in Aurel (tier 4 below shown interval). The blue frame marks the shallowest chamber acting as

1140    trap for sediment shed from above and minimizing further downward sediment transfer. B) Close-up of trapping chamber; once lateral spill-out from the chamber reaches angle of repose (ca. 30°, Allen, 1992), additional sediment from above piles up in the shaft and deeper parts of the burrow remain open. C) Sketch of fluid circulation in and around the Aurel pseudobioherm (PBH) during its formation. Lower part representing hypothetical deep structure focusing late thermogenic methane into the pseudobioherm; red dashed arrows at depth indicate hypothetical gas

1145    migration pathways from a source rock into the structure and further up to the base of the pseudobioherm. Upper part, circulation pattern of gas and fluids within the pseudobioherm; orange zigzag lines represent crustacean burrows providing a connected tube network down to the bottom of the pseudobioherm. White ellipses represent permeability tensors in the pseudobioherm and host sediment, with the principal axes of minimum and maximum permeability shown by black arrows; red arrow within the pseudobioherm marks vertical flow pattern of the gas (-charged fluids), likely

1150    with episodic escape of gas at the seafloor (red bubbles in water column).

**Appendix 1**

| Sample # | δ13C (‰ V-PDB) | δ18O (‰ V-PDB) | Description |
|---|---|---|---|
| 80 | 1.04 | -2.20 | marl |
| 2_1 | -8.55 | -2.00 | micrite |
| 2_2 | -10.16 | -1.76 | peloidal grainstone |
| 2_3 | -9.38 | -1.52 | microsparite as rim of cavity |
| 2_4 | -7.27 | -1.70 | peloidal grainstone |
| 2_5 | -8.51 | -1.55 | peloidal grainstone |
| 2_7 | -7.83 | -1.67 | Sparite 1 |
| 2_8 | 0.05 | -0.85 | Dolomite 2 |
| 2_9 | -7.49 | -1.94 | micrite |
| 3_1 | -3.30 | -1.73 | micrite |
| 3_2 | -5.84 | -1.52 | peloidal grainstone |
| 3_3 | -4.87 | -1.29 | peloidal grainstone |
| 3_4 | -5.69 | -1.62 | micrite |
| 3_5 | -5.26 | -1.39 | peloidal grainstone |
| 3_7 | -0.06 | -7.48 | Dolomite 1 |
| 3'_1 | -4.57 | -1.14 | peloidal grainstone |
| 3'_4 | -1.65 | -3.09 | Sparite 2 |
| 15_1 | -4.83 | -1.60 | micrite |
| 15_2 | -5.00 | -1.47 | peloidal grainstone |
| 20_1 | -5.28 | -1.36 | peloidal grainstone |
| 2,1_1 | -8.53 | -1.65 | microsparite as rim of cavity |
| 2,1_4 | -8.34 | -1.36 | Sparite 1 |
| 2,1_5 | -9.83 | -1.10 | Sparite 1 |
| 2,1_6 | 0.05 | -2.08 | Dolomite 1 |
| 2,1_7 | -0.27 | -0.97 | Dolomite 2 |
| 2,2_1 | -11.39 | -0.90 | microsparite as rim of cavity |
| 2,2_2 | -1.55 | -1.70 | Sparite 2 |
| 2,2_3 | -1.78 | -1.47 | Dolomite 2 |
| 3,1_1 | -2.85 | -1.62 | Dolomite 2 |
| 3,1_2 | -3.77 | -1.10 | micrite |
| 3,1_3 | -4.67 | -1.76 | peloidal grainstone |
| 1_1 | -2.42 | -1.87 | micrite |
| 84_1 | -4.61 | -1.66 | micrite |
| 84_2 | -4.61 | -1.70 | micrite |
| 84_3 | -4.47 | -1.67 | micrite |
| 84_4 | -3.99 | -2.03 | micrite |
| 84_6 | -10.42 | -1.91 | micrite |

1160

**Highlights**

- A 15 m-thick and 10 m-wide columnar carbonate pseudobioherm shows typical facies architecture.
- The pseudobioherm is characterized by a dense network of subvertical and horizontal crustacean burrows.
- About 200–500 burrows m² remained long time open after burial of the column.
- Early cements within burrows are depleted in [13]C, indicating AOM shallow in sediment.
- The open burrow network promoted vertical growth of the methane seep carbonate pseudobioherm.

1155